# Continuous resin refilling and hydrogen bond synergistically assisted 3D structural color printing

Yu Zhang[1,2], Lidian Zhang[1,2], Chengqi Zhang[3], Jingxia Wang [2,4], Junchao Liu[4], Changqing Ye[5], Zhichao Dong [2,4], Lei Wu [1,2] ✉ & Yanlin Song [1,2] ✉

3D photonic crystals (PCs) have attracted extensive attention due to their unique optical properties. However, fabricating 3D PCs structure by 3D printing colloidal particles is limited by control of assembly under a fast-printing speed. Here, we employ continuous digital light processing (DLP) 3D printing strategy with hydrogen bonds assisted colloidal inks for fabricating well-assembled 3D PCs structures. Stable dispersion of colloidal particles inside UV-curable system induced by hydrogen bonding and suction force induced by continuous curing manner cooperatively realize the simultaneous macroscopic printing and microscopic particle assembly, which endows volumetric color property. Structural color can be well regulated by controlling the particle diameter and printing speed, through which various complex 3D structures with desired structural color distribution and optical light-guide properties are acquired. This 3D color construction approach shows great potential in customized jewelry accessories, decoration and optical device preparation, and will innovate the development of structural color.

Structural color deriving from the interaction between light and periodic micro- or nanostructures has attracted increasing interest due to its eco-friendly, fade-resistant and low-toxic properties compared with the pigmentary color[1–5], which has shown great potentials in diverse applications, such as sensors[6–8], displays[9–11], anti-counterfeiting[12–14], optical devices[15–17], and so on. The 3D PCs structure can realize the control of optical path and light properties, such as polarization, phase or amplitude regulation, and results in enhanced or new optical properties that are not observed in naturally occurring materials[18–21], which has aroused intensive investigations. 3D printing can fabricate arbitrary geometries without the template prefabrication, etching, or masking required in the traditional process[22,23], and have been employed to construct complex 3D photonic structures (Supplementary Table 1). In detail, inkjet printing[24–26], direct ink writing[27–29], and

fused deposition modeling[30] have been demonstrated to prepare patterned structural color from various building blocks including colloidal particles[31,32], liquid crystals[33,34], or block copolymers[35,36]. However, the construction freedom in 3D is low and limited to discrete and plane shapes. Moreover, the cumbersome equilibrium coloration process and the weak volumetric structural color impede their wide applications. Two-photon polymerization printing of lattice structures[37] has been proved to realize the 3D color fabrication, but is compromised markedly by the print dimension and productivity. Though discontinuous 3D printing process[38] solves the problem in rapid preparation of 3D PCs structure, the rough surface and the poor fidelity still impede their applications in 3D optical devices. Therefore, the deterministic and large-scale fabrication of 3D structural color with smooth sidewall and brilliant volumetric color property through a

[1]Key Laboratory of Green Printing, Beijing National Laboratory for Molecular Sciences (BNLMS), Institute of Chemistry, Chinese Academy of Sciences, Beijing 100190, P. R. China. [2]University of Chinese Academy of Sciences, Beijing 100049, P. R. China. [3]Beihang University, Beijing 100191, P. R. China. [4]Key Laboratory of Bio-inspired Materials and Interfacial Science, Technical Institute of Physics and Chemistry, Chinese Academy of Sciences, Beijing 100190, P. R. China. [5]School of Chemistry, Biology and Materials Engineering, Suzhou University of Science and Technology, Suzhou 215009, P. R. China. ✉e-mail: wulei1989@iccas.ac.cn; ylsong@iccas.ac.cn

simple and facile method remains a challenge. Among various structural color materials to produce photonic structure, the assembly of PCs from monodispersed colloidal nanoparticle suspensions has been intensively explored due to the effective fabrication strategy and facile modification versatility[39–41]. The color generation mechanism is based on the crystalline ordering of colloidal particles in the dispersion, which requires precise temperature and humidity control. In addition, the equilibration time should be sufficient enough, which is contrary to the relatively rapid manufacturing speed of 3D printing. Though adding UV-curable additives[28,42,43], volatile solvent[44], or other polymer additives[45] into the colloidal particle solution can adapt the dispersion to the 3D printing process, it is still hard to realize 3D volumetric structural color. The uniform distribution of colloidal particles inside the printable materials during and after the printing process is critical for 3D structural color generation. Here we directly employ continuous DLP 3D printing strategy with hydrogen bonds assisted colloidal inks for the fabrication of 3D PCs structures with vibrant structural color and volumetric color property. UV-curable structural color ink is designed, in which the hydrogen bonds formed between poly(styrene-methyl methacrylate-acrylic acid) latex particles (PS latex particles) and photocurable monomers guarantee the stable dispersion of latex particles and the uniform distribution of monomers around the latex particle surface. The suction force induced by the continuous curing manner ensures the ink constantly refill inward to realize the confined assembly inside each cured layer, which enables the simultaneous macroscopic printing and microscopic particle assembly. Close-packed hexagonal assembly of latex particles is achieved inside the whole 3D structure after the subsequent evaporation-induced shrinking, resulting in brilliant structural color in the visible range. The structural color can be well controlled by regulating the particle diameter and the printing speed. Accordingly, various complex 3D structures of single or multi-structural colors can be printed through segmental printing with desired structural color distribution and optical light-guide properties. Moreover, the 3D

printed PCs structures have excellent shape fidelity, high precision, and angle dependence, which is of great importance to innovate the manufacturing method of 3D structural color and extend the application to the construction of customized jewelry accessories, decoration, and optical device functionalization.

## Results

### Ink design and continuous DLP printing 3D structural color process

As shown in Fig. 1a, the fabrication of 3D PCs structure is performed using a self-made continuous DLP 3D printing apparatus[46–48] with bottom-up projection, which mainly comprises a supporting plate mounted on the programmable moving platform, UV-transparent photo-curing interface and UV projector. Self-made UV-curable structural color ink, which consists of the UV-curable system to realize 3D constructing, aqueous solution of PS latex particles to provide structural color and carbon black (CB) to decrease incoherent scattering, is formulated for printing (Fig. 1b). The UV-curable system here is a water-based ink prepared by mixing monomer acrylamide (AM), crosslinker poly(ethylene glycol) diacrylate (PEGDA, Mn 700) and hydrosoluble photoinitiator diphenyl(2,4,6-trimethylbenzoyl)phosphie oxide (TPO-H) modified from TPO[49] (Supplementary Figure 1), which is polymerized to polyacrylamide under UV irradiation. Stable dispersion is critical for the 3D structural color realization, otherwise phase separation will occur, which results in the micro aggregation of PS latex particles. Moreover, the distribution of micro aggregations inside the polymer skeleton is uncontrollable, where the ordering assembly cannot be ensured and the structural color will be suppressed or even disappear. Accordingly, latex particles with a hard PS core and an elastomeric PMMA/PAA shell are thus employed, on which the occurrence of the surface carboxyl groups (-COOH) functions to realize the stable distribution of particles inside the polymer skeleton even for the directly cured structure (Supplementary Figure 2). This can be ascribed to the formation of hydrogen bonds between the

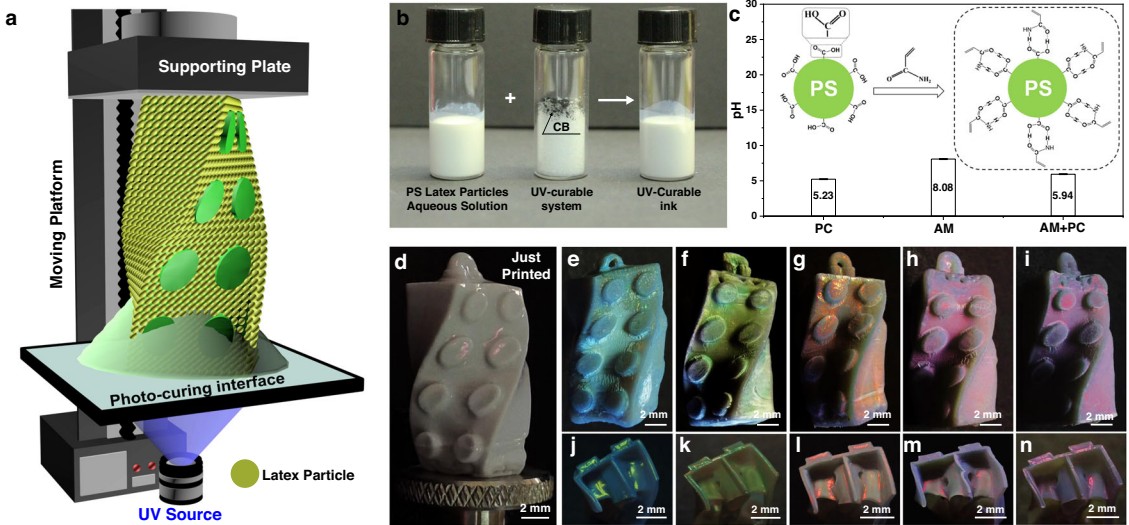

**Fig. 1 | Continuous DLP 3D printing volumetric structural color with hydrogen bonds assisted colloidal inks. a** Schematic of the continuous DLP 3D printing apparatus for fabricating 3D Lego brick structure with volumetric color property. **b** UV-curable structural color ink formation, UV-curable AM-based system mixed with aqueous dispersion solution of poly(styrene-methyl methacrylate-acrylic acid) latex particles (PS latex particles) and carbon black (CB), for 3D structuring. **c** pH characterization of pure aqueous dispersion solution of PS latex particles, pure monomer acrylamide (AM) aqueous solution, and aqueous mixture of PS latex particles with AM. Inset is the schematic diagram of hydrogen bonds formed between the -CONH₂ in monomer AM and the -COOH on the surface of PS latex

particles. The introduction of AM does not lead to an obvious change in the pH value of the aqueous dispersion solution of PS latex particles, which can ensure the stable dispersion of PS latex particles without aggregation. **d** Optical image of the just printed 3D Lego brick structure after post-washing. **e–i** Optical images of 3D Lego brick structures with different structural colors printed from PS particle diameters of 192 nm (**e**), 214 nm (**f**), 230 nm (**g**), 245 nm (**h**), and 265 nm (**i**), respectively. **j–n** Optical images of the cross-section of 3D Lego brick structures with different structural colors printed from PS particle diameters of 192 nm (**j**), 214 nm (**k**), 230 nm (**l**), 245 nm (**m**), and 265 nm (**n**), respectively.

amide group (-CONH$_2$) of monomer AM and the carboxyl groups (-COOH) of PS latex particles, which leads to the uniform bonding of monomers on the surface of PS latex particles, promotes the stable dispersion of the UV-curable structural color ink and realizes the even distribution of PS latex particles inside the printed structures.

To prove the existence of hydrogen bonds, pH characterization is first investigated as the surface -COOH of the PS latex particle is susceptible to the acidity or alkalinity of the aqueous environment, which will influence the formation of hydrogen bonds and determine the stability of the latex particles dispersed in the aqueous solution. As shown in Fig. 1c, the pH value of the pure aqueous solution of PS latex particles is 5.23, indicating that the carboxyl group on the surface of PS latex particle occurs in the form of COOH rather than COO$^-$ [50]. The pH of pure monomer AM solution is 8.08 due to its alkaline nature. While after adding AM in the PS latex particle solution, the pH increases slightly from 5.23 to 5.94, which still maintains a weak acid environment. Therefore, the introduction of monomer AM does not significantly influence the existence of carboxyl group and can form hydrogen bonds with the PS latex particles[51] (Fig. 1c inset). Furthermore, to prove the function of hydrogen bonds in suppressing phase separation, another UV-curable system with PEGDA as monomer with the other components unchanged is prepared for comparison. Without the group that can form hydrogen bonds, phase separation occurs with the PS micro-aggregates randomly distributing inside the polymer skeleton (Supplementary Figure 3), which further proves the function of hydrogen bonds in ensuring the stable dispersion of PS latex particles without aggregation and the uniform distribution of monomers around the PS latex particle surface. In addition, zeta potential of the UV-curable structural color ink ($-30.77 \pm 0.21$ mV) does not change significantly comparing to the PS latex particles in the pure aqueous solution ($-36.07 \pm 0.75$ mV), which ensures a high suspension stability for 3D printing. Therefore, UV-curable structural color inks containing PS latex particles with diameters of 192 nm, 214 nm, 230 nm, 245 nm, and 265 nm (Supplementary Figure 4), whose stopbands of hexagonal ordered assembly are in the visible range, are designed and employed for further printing.

In a specific experiment, a droplet of UV-curable structural color ink is deposited on the photo-curing interface before printing. By continuously projecting sequences of light patterns at the upper surface of the photo-curing interface and simultaneously elevating the supporting plate at a set speed, 3D Lego brick structure is finally obtained (Supplementary Movie 1). Due to the existence of water in the polymer skeleton, the just printed 3D Lego brick structure from 214 nm PS latex particles displays red structural color after post-washing (Fig. 1d). After the water evaporation process, the 3D printed Lego brick structure shrinks in volume and displays the blue-shifted structural color (Fig. 1f). In addition, the cross-section of the printed structure also displays vivid structural color (Fig. 1k), which demonstrates that the printed structure has volumetric color property. The preparation of complex 3D Lego bricks with different brilliant structural color and volumetric color property can also be realized for PS latex particles with different particle diameters (Fig. 1e–n). Therefore, the mechanism of the generation of structural color is further investigated below.

## The mechanism of structural color generation
To study the mechanism of structural color generation, slab structure (10 mm in length and height, 500 μm in width) is selected as the representative. As shown in Fig. 2a, with the continuous elevation of the supporting plate along with the ink refilling, slab structure can be 3D printed accordingly. During the water evaporation process, the color of the printed slab structure gradually blue-shifts from red of the partially evaporated structure (Fig. 2b), and finally turns to green-yellow of the completely evaporated structure (Fig. 2c). Simultaneously, the volume of the printed sample shrinks uniformly along

with the color change. Therefore, the distribution and motion tendency of PS latex particles and polymer skeleton during the continuous printing process and the evaporation process are systematically characterized through scanning electron microscope (SEM) to investigate the structural color generation mechanism. In detail, the surface and cross-section morphologies of the printed samples at different states including the just printed (Fig. 2d), after partial evaporation (Fig. 2e), and complete evaporation (Fig. 2f) are characterized. For the sample containing water, freeze-drying is conducted to investigate the actual distribution of polymer skeleton and PS latex particles. As shown in Fig. 2d of the just printed sample, the PS latex particles are orientationally assembled, which is in consistence with the ink refilling direction during the continuous printing process (yellow arrows in Fig. 2d), and the polymer skeleton contacts with the PS latex particles in the form of the filament. For the partially evaporated sample, the length of the polymer filaments and the PS latex particle spacing accordingly decrease along with the water evaporation (Fig. 2e). In addition, the contact sites of filaments are uniformly distributed around each PS latex particle, which further proves the uniform distribution of monomers as well as the polymer skeleton around the PS latex particle through hydrogen bonds. Consequently, the reduced distance between the PS latex particles makes the stopband of the printed part locate in the visible range, which displays red structural color. After water complete evaporation, the PS latex particles form a close-packed hexagonal arrangement and the polymer chains uniformly intersperse the gaps among the PS latex particles on all side surfaces and throughout the entire cross sections (Fig. 2f, Supplementary Figure 5), which endows a blue-shifted vivid structural color compared to the partially evaporated sample. In addition, the morphology of the polymer skeleton is characterized through selectively removing the PS latex particles with toluene. As displayed in Fig. 2g, the polymer skeleton surface displays the arrayed semi-holes morphology with hexagonal assembly, and the cross-section shows arrayed cilia morphology that replicates the vertical gaps among the close-packed particles, above of which further confirm the uniform distribution of PS latex particles and the uniform interspersing of the polymer skeleton among PS latex particles in this system through hydrogen bonding. In addition, the time-sequence color change of the printed 3D bear structures further proves the simultaneous macroscopic printing and microscopic particle assembly mechanism, as shown in Supplementary Movie 2.

Thus, the brilliant structural color is generated from the uniform dispersion and hexagonal assembly of PS latex particles inside the polymer skeleton, which can be attributed to the synergistic effect of ink filling during the continuous printing process and the hydrogen bonds formation to maintain a uniform dispersion of the PS latex particles both inside the ink and the cured polymer skeleton. In detail, as displayed in Fig. 2h, during the continuous printing process, the cured structure is always immersed inside the uncured ink, creating a low-pressure zone under the previously printed layers, which induces the generation of suction force when the supporting plate is continuously moved up[52]. The suction force can lead to the ink continuously refill inward between the cured structure and the curing interface, where the PS latex particles are solidified and confined assembled with a certain orientation inside the polymer skeleton with the assistance of hydrogen bonds (Fig. 2i). After printing and during the solvent evaporation process, the existence of hydrogen bonds guarantees the uniform distribution of PS latex particles inside the shrinking polymer skeleton along with the water loss (orange dotted box in Fig. 2j) until complete evaporation, where crystalline and structural color finally occurs. While for the same slab structure printed from discontinuous printing process (z-axis stepping length of 20 μm and staying time of 6 s for each layer), multiple separate steps are needed for printing one single layer[53] (Supplementary Figure 6a), where no low-pressure zone and suction force are generated, leading

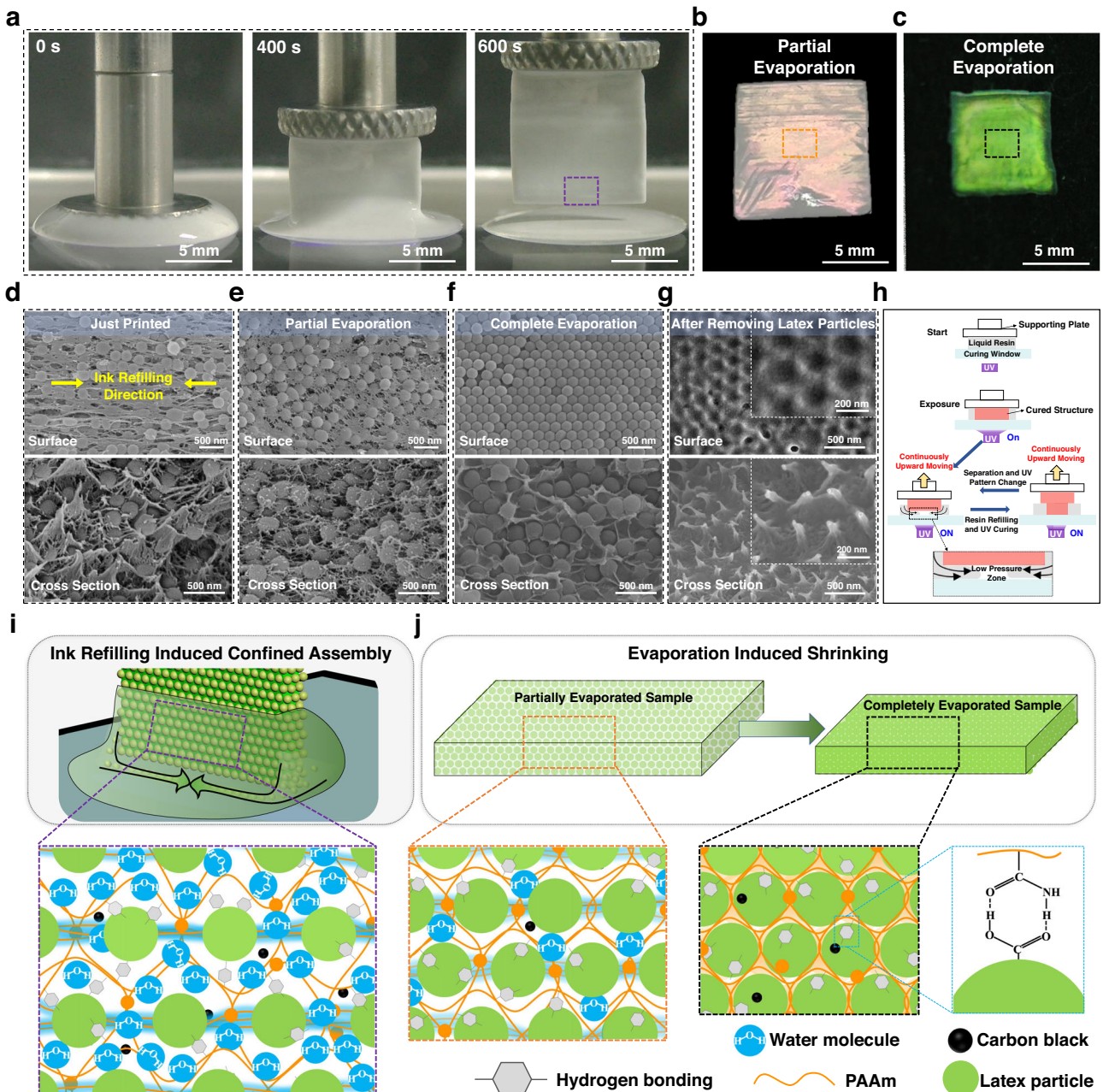

**Fig. 2 | Mechanism of 3D structural color generation. a** Time-sequence optical captures of the continuous DLP 3D printing process of the slab structure using the UV-curable structural color ink, where the PS particle diameter is 214 nm. **b** Optical image of the 3D printed slab structure after partial evaporation, displaying red structural color. **c** Optical image of the 3D printed slab structure after complete evaporation, displaying green-yellow structural color. **d**–**f** Surface and cross-sectional SEM images of the just printed (**d**), partially evaporated (**e**), and completely evaporated 3D structure (**f**). The yellow arrows in (**d**) represent the ink refilling direction during the continuous printing process. **g** Surface and cross-sectional SEM images of the completely evaporated 3D structure after selectively removing the PS latex particles. **h** Scheme of the continuous DLP 3D printing process. Suction force is generated from the low-pressure zone as the cured structure is always immersed inside the uncured ink. **i** Schematic illustration of the ink refilling induced confined assembly during the continuous printing process. Arrows represent the ink refilling direction inside the droplet. The purple dotted box indicates the distribution of polymer skeleton and PS latex particles inside the just printed 3D printed structure, which contains a large proportion of water and a far distance between PS latex particles. **j** Schematic illustration of the evaporation-induced shrinking process. The orange dotted box indicates the distribution of polymer skeleton and PS latex particles inside the partially evaporated 3D printed structure. The black dotted box indicates the distribution of polymer skeleton and PS latex particles inside the completely evaporated 3D printed structure, where the PS latex particles are close-packed with polymer chains evenly interspersing the gaps among them through the hydrogen bonds. The blue dotted box is the scheme of the hydrogen bonds between the -CONH$_2$ of AM and the -COOH of PS latex particles.

to uncontrollable assembly of PS latex particles inside the polymer skeleton and unobvious structural color (Supplementary Figure 6b–d). Along with the amorphous assembly of the directly cured structure (Supplementary Figure 2) which also excludes the suction force, it can be concluded that the suction force induced by the continuous DLP 3D printing and the continuous resin refilling determine the realization of

crystalline assembly, and the crystalline occurs after complete evaporation. Under the synergistic effect of hydrogen bonds (blue dotted box in Fig. 2j) and ink refilling during the whole continuous printing process, the simultaneous macroscopic printing and microscopic particle assembly can be realized, the printed structure is thus imparted with hexagonal assembly and vivid volumetric color

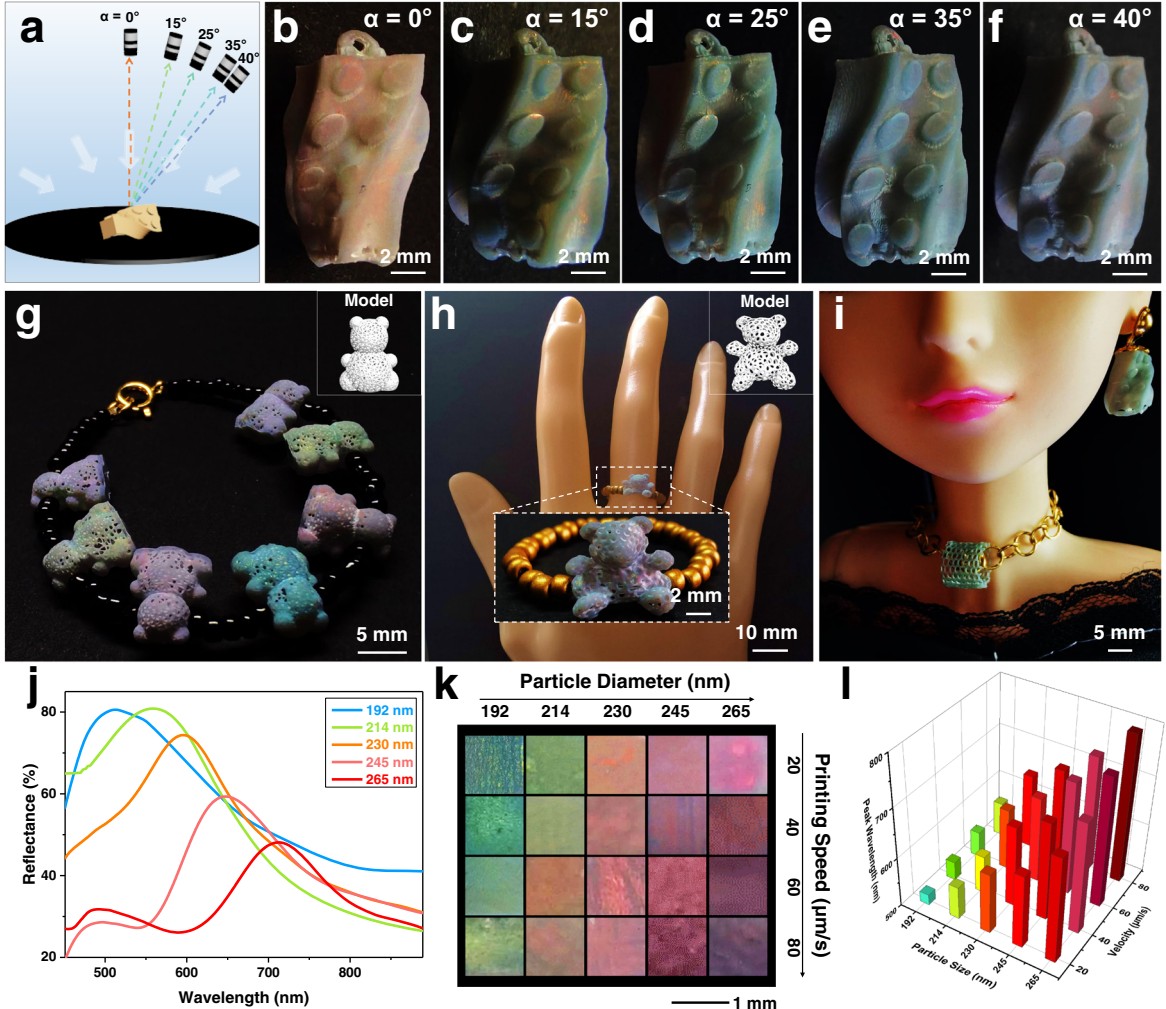

**Fig. 3 | Structural color regulation. a** Scheme of structural color observation of Lego brick structure from the different viewing angles (α) under diffusive lighting condition. **b–f** Optical images of the Lego brick structure with viewing angles (α) from 0° (**b**) to 15° (**c**), 25° (**d**), 35° (**e**), and 40° (**f**) under diffusive lighting condition. The Lego brick structure is printed with PS particle diameter of 230 nm. **g** Optical image of the customized bracelet structure composed of 3D printed bear structures of different structural colors and black plastic beads. The bears from the left to right are printed with PS particle diameters of 265 nm, 214 nm, 245 nm, 192 nm, 230 nm, 214 nm, and 265 nm, respectively. The inset of the solid white box on the upper right is the 3D bear model. **h** Optical image of a hand model wearing the customized ring structure, which is fabricated by connecting the 3D printed bear structure with golden plastic beads. The insets of the white dotted box in the middle and the solid white box on the upper right are the optical images of the magnified ring structure and the 3D bear model, respectively. **i** Optical image of a Barbie doll wearing the customized earring and necklace structures. The necklace structure is a designed cylinder grid structure printed from 214 nm PS latex particle with decorative chain (metal rings) passing through, and the earring structure is the Lego Brick structure printed from 214 nm PS latex particle with its top ring connected with an ear stud (metal). **j** Reflectance spectra of the 3D structures printed from PS latex particles with different diameters. **k** Composite optical images of the 3D structures printed with varying PS latex particle diameters and printing speeds. Scale bar: 1 mm. **l** Phase diagram of the characteristic peak wavelength versus PS latex particle diameters and printing speeds.

property (black dotted box in Fig. 2j). Furthermore, the mechanism of the 3D printed PCs structures is versatile to PS latex particles with different diameters (Supplementary Figure 7).

## Systematic control of the structural color of 3D printed structure

Having elaborated the mechanism of the structural color generation, the factors that influence the structural color of the 3D printed structure including the viewing angle, printing parameters and particle diameter are further investigated. The viewing angle (α) influenced color change is first investigated (Fig. 3a). As shown in Fig. 3b, the 3D Lego brick structure printed with PS particle diameter of 230 nm shows the orange color at normal viewing angle (α = 0°) under diffuse lighting of sunlight. When the viewing angle increased from 0° to 40°, the color blue-shifts from orange to green-yellow and finally to bluish violet, as shown in Fig. 3b–f, which exhibits the angle-dependent

structural color and can further prove the hexagonal assembly of the 3D printed PCs structure. When employing the printed Lego Brick structure as earring structure, it displays different colors viewing from different angles (Supplementary Figure 8), which can prove the capability of 3D printed PCs structures for application as jewelry accessories with color diversities. In addition to the earring structure printing, a variety of gorgeous 3D PCs structures can be facilely constructed including the bracelet structure (Fig. 3g), the ring structure (Fig. 3h), and the necklace structure (Fig. 3i), all of which exhibit vivid structural colors, demonstrating their feasibility in printing customized accessory designs. Then, the influence of printing parameters including the slicing thickness, printing angle and printing speed on the structural color of the printed structures are investigated. Ascribing from the continuous printing mode and the synergistic color generation mechanism, the influence of slicing thickness and printing angle on the wavelength of the stopband can be ignored

(Supplementary Figure 9). Without PS latex particles, no stopband can be observed on the reflectance spectra of the pure UV-curable system (Supplementary Figure 10). Therefore, the structure color stopbands of 3D printed structures varied via the modulation of the printing speeds are further investigated along with the PS latex particle diameter regulation.

As displayed in Fig. 3j and Supplementary Figure 11, with a fixed printing speed of 20 μm/s, the structural color of the printed slab structure can be regulated from cyan, green-yellow, orange, red-orange to red through increasing the diameter of the PS latex particle from 192 nm, 214 nm, 230 nm, 245 nm to 265 nm without changing other ink components. In addition, comparing with the assembly of pure PS latex particles, the interspersion of polymer skeletons among the PS latex particles increases the particle spacing, making the wavelength of the stopband red-shifted (Supplementary Figure 12), which are consistent with the calculated results (Supplementary Figure 13) by the Bragg's law[54,55]:

$$\lambda_{max} = 2d_{111}n_{eff}\sin\theta \tag{1}$$

Where $\lambda_{max}$ is the wavelength of the peak position, $d_{111}$ is the lattice spacing, $n_{eff}$ is the effective refractive index of the composite PCs, $\theta$ is the incidence angle (90° in the experiment). $d_{111}$ and $n_{eff}$ can be expressed as:

$$d_{111} = \sqrt{2/3}D \tag{2}$$

$$n_{eff}^2 = fn_0^2 + (1-f)n_c^2 \tag{3}$$

Where $D$ is the diameter of latex particles, $f$ is the filling factor, $n_0$ (1.60) and $n_c$ (1.36) are the refractive index of PS latex particles and hydrogel, respectively. Along with the regulation of printing speed, as shown in Fig. 3k and Supplementary Figure 14, a series of printed slab structures with different structural colors and reflectance spectra are obtained by varying the printing speeds from 20 μm/s to 40 μm/s, 60 μm/s, and 80 μm/s with five PS latex particle diameters (192 nm, 214 nm, 230 nm, 245 nm, and 265 nm). It can be found that with the increase in printing speed for the same PS latex particle diameter, the positions of the stopband red-shift and the structures exhibit corresponding color change. As different printing speeds require different UV intensities and exposure times of one layer, which leads to different printing times and different ratios of PS particles for the same 3D structure. In detail, higher printing speed under higher UV intensity will lead to larger dry volume of polymer skeleton (Supplementary Figure 15) and a smaller ratio of PS latex particles solidified inside the skeleton, resulting in a larger particle spacing (Supplementary Figure 16) and a red-shifted structural color. Therefore, with cooperative regulation of the PS latex particle diameters and printing speeds, structural colors covering the visible range can be acquired according to the phase diagram that maps the PS latex particle diameters, printing speed and the characteristic peak wavelength in Fig. 3l. In detail, the position of the characteristic peak blue-shifts with the decreasing of the PS latex particle diameters, while red-shifts with the increasing of the printing speeds. Thus, the phase diagram can serve as a guidance for selectively printing structure with a desired color.

**Structural color control in 3D through printing manipulation**

Encouraged by the generation of brilliant structural color and volumetric color property, we further demonstrate the versatility of this strategy in printing more complex 3D structures on-demand with high freedom of morphology and structural color control in three-dimensional. Besides the Lego brick and slab structures, 3D pyramid-like, cylinder grid, ring, gyroid, and box with twisted internal structures (Supplementary Figure 17, Supplementary Movie 3) can be

continuously printed with desired structural colors. In addition, complex multi-structural colors 3D structure can also be acquired through the assembly of different parts with single structure color or direct printing. As displayed in Supplementary Figure 18, the fabrication of the detachable and combinable composite structure can be achieved through printing the three components separately, all of which have different and vivid structural colors. For direct printing, to integrate the multi-structural colors into one single structure, printing is conducted through programmatic control of the amount and latex particle diameter of inks for different segments, and the handover of each segment. As shown in Fig. 4a, the koi fish model is divided into different amounts of segments, with each segment sliced and projected separately and sequentially. By controlling the amount of required ink according to the volume of each segment, different segments are printed in sequence with the exhaustion of ink, and the printing of the following segment starts from adding corresponding amount of ink containing different PS latex particles. The structure that has been printed is not moved or removed from the supporting plate during the whole printing process. The printing fidelity and precision of the segmental printing method are further systematically investigated. The printed koi fish structure with 4-segment structural colors is shown in Fig. 4b, which is an accurate reproduction of the original model (Fig. 4b inset) and the interface between adjacent segments is not optically obvious due to the programmatic control of the printing process. In addition, as shown in Fig. 4c–f, the cross-sectional optical images of I, III, V, and VII in Fig. 4b have vivid and single structural color, which demonstrates that the segmental printing method will not influence the structural color purity of parts printed from the UV-curable structural color ink containing different colloidal particle diameters.

The surface morphology of the printed koi fish structure is then characterized, as displayed in Fig. 4g–j, where the structure details of head (Fig. 4g), gill (Fig. 4h), scale (Fig. 4i), and fin (Fig. 4j) are clearly visible. Besides, the interface between the two neighboring segments is connected by the polymer skeleton and the gap dimension is tiny (Fig. 4k), which can be ascribed to the different structural color inks employed containing the same UV-curable system. The internal morphology characterized through micro-computed tomography (Micro-CT) as displayed in Fig. 4l, m, the printed structure has excellent fidelity and high precision, which further proves the controllability and structural integrity of this method in printing 3D structure with multi-structural colors. Reflectance spectra are also characterized on different segments and interfaces to illustrate the influence of segmental printing on the structural color distribution and transition of different segments. As displayed in Fig. 4n, the wavelengths of stopbands on the four independent segments (I, III, V, and VII) are the same as the structure printed with single structural color, while the stopband wavelengths of the interface (II, IV, and VI) are between the two adjacent segments, indicating the feasibility of the segmental printing method for the fabrication of structure with distributed structural color. Thus, koi fish structures with 1-segment, 2-segment, 3-segment, and 4-segment structural colors are printed accordingly. Through decorating the 3D printed single- and multi-structural color fishes with waterweed structures, the hyperrealistic 3D picture of colorful fish with oriental artistic conception can be acquired (Fig. 4o). The distinct structural colors and stereoscopic effect makes the image lifelike and full of vitality, proving its potential in art creation and decoration.

**Optical light-guides with color and pattern selectivity**

Being capable of continuous 3D printing structures with high surface finish and volumetric color, the application in optical device preparation can thus be realized. Optical light-guide structures have attracted great attention for their ability to manipulate and transport light in a controlled manner, whose property relies on the medium morphology[56–60]. Based on the 3D printing induced coloration

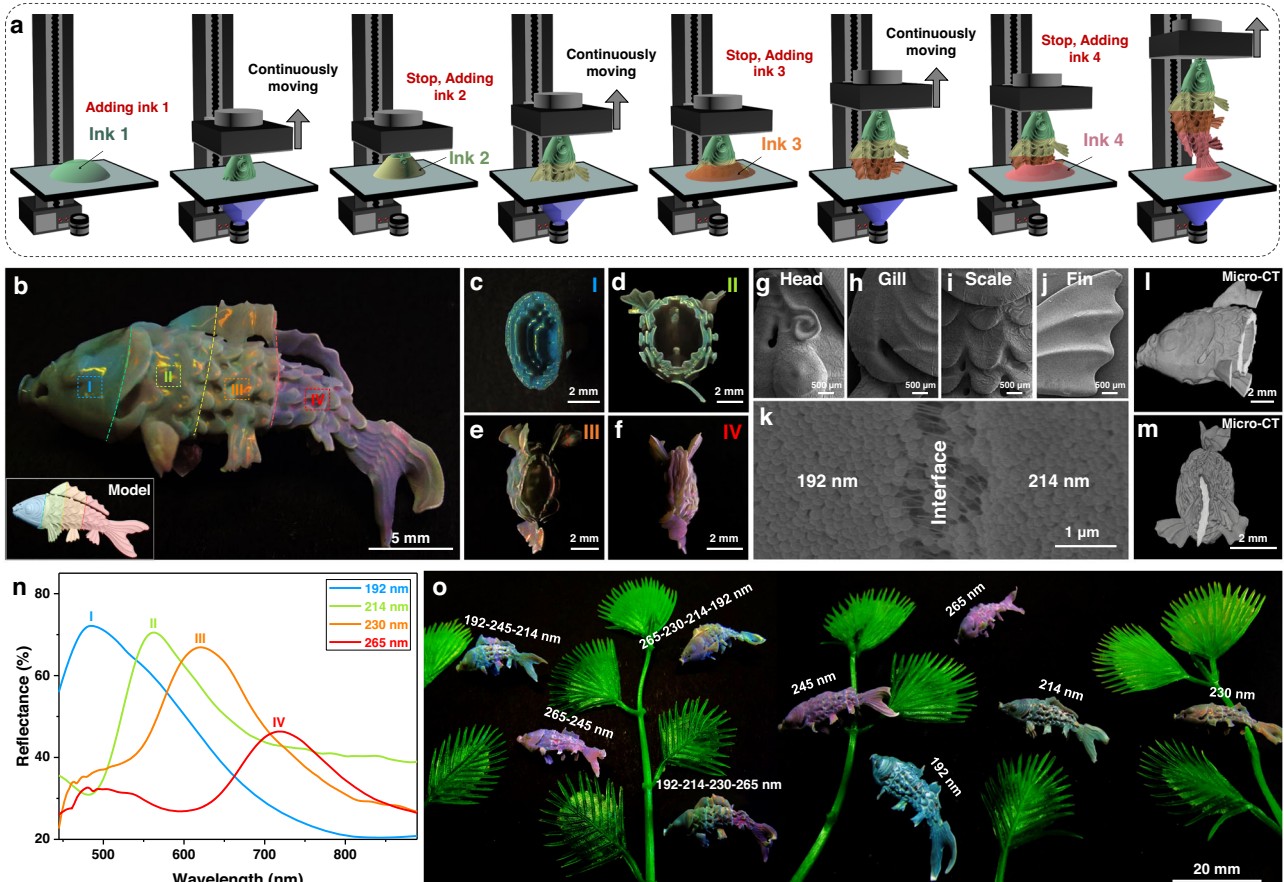

**Fig. 4 | High-precision 3D printing colorful koi fish structures. a** Scheme of the preparation process of 3D koi fish structure with multi-structural colors. **b** Optical image of the multi-structural color koi fish continuously printed with four different PS latex particle diameters. The dashed curves represent the interface of two segments printed with different PS latex particle diameters. I–IV are the segments printed from the PS latex particle diameters of 192 nm, 214 nm, 230 nm, and 265 nm, respectively. The inset on the bottom left is the 3D model of koi fish. **c**–**f** Cross-sectional optical images of I (**c**), II (**d**), III (**e**), and IV (**f**) in (**b**), respectively.

**g**–**j** SEM images of the head (**g**), gill (**h**), scale (**i**), and fin (**j**) of the 3D printed koi fish structure. **k** Surface SEM image of the interface between the segments I and II in (**b**), the two sides of the interface are printed from the PS latex particle diameters of 192 nm and 214 nm, respectively. **l, m** Cross-sectional Micro-CT images of the internal (**l**) and tail (**m**) of the 3D printed koi fish structure, indicating that the printed structure has excellent fidelity and high precision. **n** Reflectance spectra of the I–IV segments in (**b**). **o** Optical image of the 3D printed koi fishes with single or multi-structural colors. Waterweed structures are plastic decorations.

mechanism, hollow cylinder-shaped tubes with smooth inner and outer surfaces, low optical loss and color selectivity can be continuously 3D printed from PS latex particles with different diameters (Fig. 5a–e). In detail, when the white light is introduced into one opening of the structure, single-colored light corresponding to the photonic stopbands of PS latex is exported from the other opening (Fig. 5f), exhibiting the frequency selective light-guide property. In addition, the optical loss is low and is basically linearly cylinder length related (Fig. 5g–l, Supplementary Table 2), where the optical loss coefficient is ~4.57 ± 1.05 dB/cm and is comparable to the loss of light traveling the same distance in air (~3.28 ± 0.14 dB/cm). While for the hollow cylinder-shaped tube structures printed from the discontinuous printing method based on the same model (Supplementary Figure 19a–c), they have poor surface smoothness and obvious step structures. Moreover, the printing stability decreases as the printing proceeds, leading to size deviation with narrow holes and limited length preparation, which results in the increased optical loss (the optical loss coefficient is ~6.87 ± 1.07 dB/cm) and distorted output light pattern (Supplementary Figure 19d, e).

In addition, the output light pattern can also be manipulated based on the structural controllability. Triangular (Fig. 5m), square (Fig. 5n), pentagonal (Fig. 5o) and flower-shaped (Fig. 5p) colored light patterns can be output through the preparation of cylinder structures with different cross-section patterns. Besides the linear light guidance,

it is also capable to change the output direction of light from 30° to 90° comparing with the input direction through continuous printing bended structures, as shown in Fig. 5q–u. Furthermore, interlapping multiple guided lights can generate new color of light, and the rest non-overlapping areas retain their original color (Fig. 5v–x), which further confirms the morphology and color controllability of our method and its assurance in optical application. Thus, the printing induced coloration mechanism and the feasibility in PCs structure printing is advantageous not only in fine 3D structure fabrication but also in optical device functionalization.

In conclusion, we have demonstrated a facile 3D PCs structure fabrication approach with hydrogen bonds assisted colloidal inks through the continuous DLP 3D printing technology for the first time. The synergistic effect of hydrogen bonds induced uniform dispersion and the continuous curing manner induced suction force enables the confined assembly inside each cured layer, realizing the simultaneous printing and assembly, and rendering the 3D PCs structures with volumetric color property. The brilliant structural color resulting from the ordered assembly can be finely regulated by the particle diameter and the printing speed. Various complex 3D structures with desired single or multi-structural colors are fabricated via segmental printing. The successful printing of optical light-guide structures with smooth inner and outer surfaces, low optical loss and color selectivity further demonstrates the advantage of our printing induced coloration

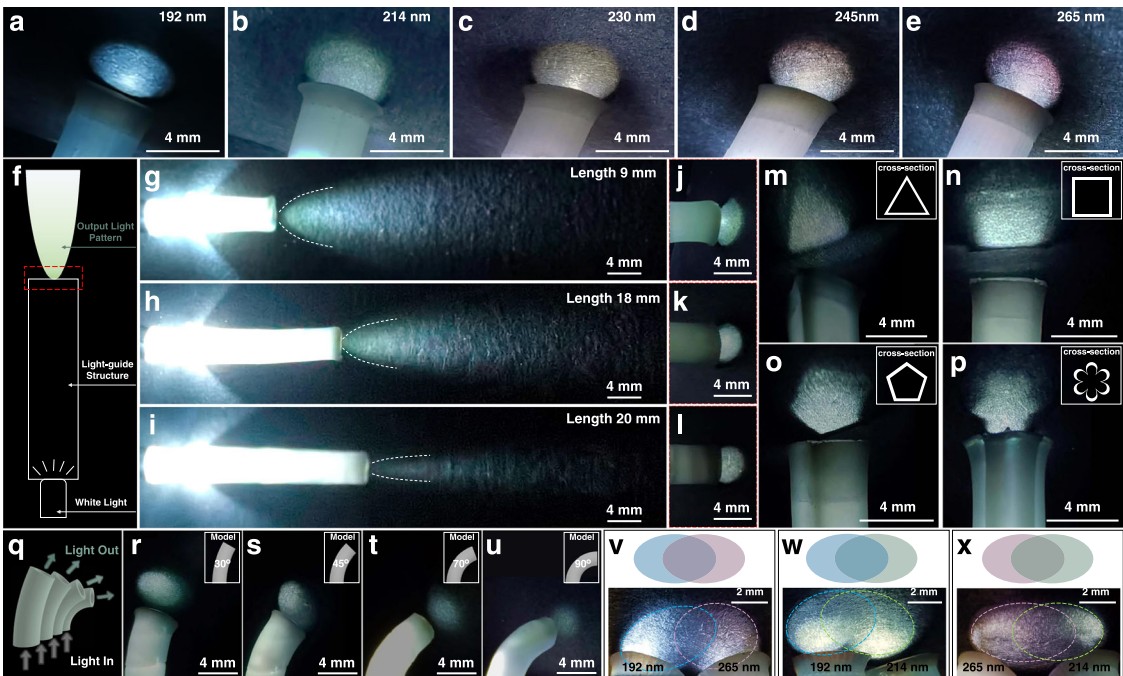

**Fig. 5 | Continuous DLP 3D printed PCs structures as color and pattern selective optical light-guide tubes. a–e** Optical images of hollow cylinder-shaped tubes with different frequency selective light-guide property printed from PS particles with diameters of 192 nm (**a**), 214 nm (**b**), 230 nm (**c**), 245 nm (**d**), and 265 nm (**e**), respectively. **f** Schematic illustration of the length-dependent optical loss characterization. **g–l** Optical images of light-guide performance of hollow cylinder-shaped tubes with lengths of 9 mm (**g, j**), 18 mm (**h, k**) and 20 mm (**i, l**), respectively. **j–l** are the optical images of output light morphology on a black background from different lengths. **m–p** Optical images of output light patterns with triangle (**m**), square (**n**), pentagon (**o**), and flower (**p**) morphologies, respectively. The insets are cross-sectional morphology of corresponding tubes. **q** Schematic illustration of light direction guidance through bended structures. **r–u** Optical images of bended optical light-guide structures with bending angle of 30° (**r**), 45° (**s**), 70° (**t**), and 90° (**u**), respectively. The insets are the corresponding models of optical light-guide structures. **v–x** Optical images of the light color regulation through overlapping lights guided from two light-guide tubes. The two light-guide tubes are printed from PS particles with diameters of 192 nm and 265 nm (**v**), 192 nm and 214 nm (**w**), and 265 nm and 214 nm (**x**), respectively.

mechanism and the feasibility in the fabrication of high-fidelity and high-precision 3D PCs structures, as well as in boosting the functionalization of optical devices.

## Methods

### Materials

The synthetic aqueous dispersion solutions of PS latex particles with the diameter of 192 nm, 214 nm, 230 nm, 245 nm, and 265 nm are used, whose solid content are 24.23 wt%, 24.10 wt%, 19.89 wt%, 22.54 wt%, and 22.70 wt%, respectively. Methyl methacrylate (MMA), acrylic acid (AA), styrene (St), ammonium bicarbonate, ammonium peroxydisulfate (APS), sodium dodecyl benzene sulfonate (SDBS), AM, PEGDA (Mn 700), sodium dodecyl sulfate (SDS), polyvinylpyrrolidone (PVP, Mn 40000), isopropyl alcohol (IPA), and n-butyl acetate (n-BuAc) are purchased from Sigma-Aldrich. TPO is purchased from Adamas. CB (granularity <90 nm) is purchased from Innochem. All chemical reagents are directly used as received without further purification.

### Synthesis of aqueous dispersion solutions of PS latex particles

Aqueous dispersion solutions of PS latex particles are synthesized via the previously reported method[61]. In detail, the reaction mixture is obtained by adding monomer MMA (10 mmol), monomer AA (13.89 mmol), monomer St (182.60 mmol), buffer reagent ammonium bicarbonate (6.3 mmol) and surfactant SDBS (mass ratio to monomer: 1:0.025–0.060) in the deionized water (100 ml), which is carried out at 70 °C for 30 min. Then, the aqueous dispersion solutions of PS latex particles are obtained by adding APS (2.32 mmol) to the aforementioned reaction mixture for mechanically stirring 10 h at 80 °C. The resulting aqueous dispersion solutions of PS latex particles are directly used without purification.

### Synthesis of hydrosoluble photoinitiator TPO-H

The water-compatible photoinitiator TPO-H is synthesized by modifying the commercial photoinitiator TPO with water-soluble components through the microemulsion method. In detail, TPO (1.7 wt%) is firstly dissolved in the oil phase, which is comprised of the mixture of volatile solvent n-BuAc (22.3 wt%), cosolvent IPA (21 wt%), surfactant SDS (7.5 wt%) and crystallization inhibitor PVP (7.5 wt%), for mechanically stirring 12 h at room temperature. Then, the oil-in-water microemulsion is obtained by mixing the above oil phase with deionized water (40 wt%). After mechanically stirring at room temperature for 12 h till turning into clear yellowish, TPO is wrapped inside the microemulsion with water as the continuous phase. Finally, the microemulsion is frozen in liquid nitrogen for 20 min and then lyophilized at the temperature of −47 ± 3° and vacuum degree of ~0.35 mbar for 24 h to white powder, i.e., TPO-H. The resulting TPO-H powder is water-soluble and is composed of 10 wt% TPO, 45 wt% SDS, and 45 wt% PVP, which is sealed and stored in a brown bottle before use.

### Ink preparation

The designed UV-curable structural color ink is prepared at room temperature by mixing the monomer AM, water-compatible photoinitiator TPO-H, crosslinker PEGDA (Mn 700), aqueous dispersion solution of PS latex particles, and additive CB. In detail, the mass ratio of PS particles to AM, TPO-H to AM, PEGDA to AM, and CB to AM are 1:2, 1:10, 1:20, and 1:33, respectively. The obtained ink is stored in the dark and generally used within one week of preparation.

### 3D printing apparatus

The self-made 3D printing apparatus with bottom-up UV illumination is displayed in Fig. 1a, which comprises a 405 nm LED UV projector

(PRO4500, Wintech, China), liquid resin vat with UV-transparent photo-curing interface (quartz cell with the bottom surface modified with perfluorocarbon swelled polydimethylsiloxane) and supporting plate (aluminum block) mounting on a programmable mobile platform (MC600, Zolix Instruments Co., Ltd. China). The projection area, light intensity range, and resolution of the UV projector are 32.2 mm × 51.6 mm, 0–65 mW/cm², and 1280 × 800 pixels, respectively. The velocity range that the programmable mobile platform can provide is 1.2–100 mm/min. The light pattern sequences played by the UV projector are acquired through slicing the 3D model into a series of cross-sectional images with slicing thicknesses of 5 μm. To investigate the influence of slicing thickness on the stopband wavelength, slicing thickness of 20 μm and 50 μm are also employed.

### 3D printing post-treatment
After printing, excess ink that adhered to the printed structure and the supporting plate is removed via the washing of the structure with deionized water until the rinsed liquid becomes clear. Finally, the sample is removed from the supporting plate and left to completely dry under ambient condition.

### Characterization
The assembly of colloidal particles is characterized using scanning electron microscope (SEM, S-4800, Hitachi, Japan) at an accelerating voltage of 5.0 kV. Micro-computed tomography (Micro-CT) images are shot and reconstructed by using Micro-CT equipment (Skyscan 1272, Bruker, Germany). Freeze-drying is conducted using the lyophilizer (FD-1A-50, Boyikang, China) to obtain the hydrosoluble photoinitiator TPO-H and the lyophilized samples for SEM characterization. Optical images of the structure and real-time monitoring of the printing process are obtained by the digital camera (CU-VF100AC, JVC, Japan). Fourier transform infrared (FTIR) spectrum is measured by FTIR spectrometer (TENSOR-27, BRUKER, Germany). Reflectance spectra are measured by fiber optic spectrometer (NoVA-EX, IdeaOptics, China) through an optical microscope (BX53, OLYMPUS, Japan). The values of pH are measured using a laboratory pH meter (FE20, METTLER TOLEDO, China). Zeta potentials of PS latex particles in the pure aqueous solution and the aqueous mixture are measured by Zetasizer Nano (Nano ZS ZEN3600, Malvern Instrument, UK). Each reported zeta potential is an average of at least three independent measurements. Mechanical properties of the just printed 3D PCs structure and pure acrylamide hydrogel structure are measured by tensile testing at a drawing rate of 20 mm/min⁻¹ using universal testing machine (Instron5567, USA).

## Data availability
The authors declare that the main data supporting the findings of this study are contained within the paper. All other relevant data are available from the corresponding author upon request. Source data are provided with this paper.

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

## Acknowledgements

We acknowledge funding of the National Natural Science Foundation of China (Grant Nos. 52173051 to L.W., 51773206, 51961145102, and 91963212 to Y.S.), the Youth Innovation Promotion Association (2021031 to L.W.), the National Key R&D Program of China (Grant Nos. 2018YFA0208501 and 2018YFA0703200 to Y.S.), the K.C. Wong Education Foundation (to Y.S.) and the Beijing National Laboratory for Molecular Sciences (BNLMS-CXXM-202005 to Y.S.).

## Author contributions

L.W. and Y.S. conceived and designed the experiments. Y.Z. and L.W. performed the experiments and analyzed the data. L.Z. helped synthesize the hydrosoluble photoinitiator TPO-H. C.Y. helped synthesize the aqueous dispersion solutions of PS latex particles. C.Z. and Z.D. helped reconstruct the Micro-CT images. J.L. and J.W. helped characterize the reflectance spectra. Y.Z. wrote the original manuscript. L.W. and Y.S. helped revise it. All authors proofread the paper, made comments, and approved the manuscript.

## Competing interests

The authors declare no competing interests.
