## [Peer Review File · Nature Communications]

Continuous Resin Refilling and Hydrogen Bond Synergistically Assisted 3D Structural Color PrintingReviewers' Comments:

Reviewer #1:

Remarks to the Author:

This manuscript reports continuous DLP printing of structurally colored 3D objects using colloidal suspensions. The suspensions are composed of PS@PMMA/PAA core-shell particles in the aqueous mixture of PEGDA, AAm, and a photoinitiator. Rich carboxylic groups on the surface of particles stabilize the suspension and make hydrogen bonds with AAm, which helps the particles to be evenly distributed in the final particle-polymer composite produced by photo-polymerization of AAm and PEGDA. 3D structures are continuously produced from a large drop of the suspension by the DLP technique during UV exposure, at which particles are embedded. After evaporation of water, particles are arranged to have some order, which develops structural colors in the shrunken 3D structures. As the drop of suspension is confined by capillary force, distinct suspensions can be consecutively employed to produce multi-colors in a series for single 3D structures. This work is generally interesting. But, the manuscript lacks detailed analysis. Furthermore, the authors might miss the recent paper on 3D printing of colloidal photonic structures, J. Liao et al. *Materials Today* (2022), <https://doi.org/10.1016/j.mattod.2022.02.014>. The authors should strengthen their work by addressing the following comments:

1. The 3D printing of structurally colored objects based on the DLP technique was reported recently in *Materials Today* (2022). The authors should modify the introduction and conclusion to review the previous work and justify the novelty and advances of this work.
2. There is no discussion of how and when the particles produce ordered structures. First, the authors should analyze the internal structure of colloidal arrays to confirm whether the structures are crystalline or amorphous. Second, if the particles form crystalline arrays, the authors should study when the crystallization occurs and how the crystal orientation is determined (affected by the DLP process or drying?). It seems that the particles are not ordered in the printed structure before the evaporation. Is there any influence of the suction force and ink refilling on the crystallization process? If the structure is crystalline, it is strongly suggested to analyze the reflectance spectra with Bragg's equation.
3. It is difficult to understand what the suction force is and how the suction force plays a role in particle ordering. Is there no suction force for stepwise printing?
4. It is suggested to study why stopband position varies with printing speed.
5. How do the PS@PMMA/PAA core-shell particles maintain high suspension stability in the aqueous mixture of PEGDA, AAm, and photoinitiator despite the possible charge screening by AAm? It would be helpful to measure the zeta potentials of the particles in distilled water and the aqueous mixture.
6. The boundary between two adjacent parts with different colors is sharply defined in Figure 4? What causes a single reflectance peak, instead of two peaks, at the boundary which is positioned between the peaks of the two adjacent parts?
7. What causes the broadening of reflectance peak for 3D printed structure in Supplementary Figure 10? It is suggested to the absolute unit for all reflectance spectra. What is the gravitational deposition method?
8. What is the slicing thickness on page 10?

Reviewer #2:

Remarks to the Author:

In this manuscript, 3D photonic crystal with volumetric color was prepared by a continuous digital light processing (DLP) 3D printing method. The preparation of colloid ink and the control of the printing process was demonstrated. DLP technique with an ink composed of acrylamide and PEGDA have been widely studied to fabricate 3D biomaterials (*Adv. Healthcare. Mater.* 2020, 9, 2000156), and it is a good idea to extend this technique to the fabrication of 3D PC structures. However, some statements in this paper could not be well supported/explained by the discussion. Therefore, a major revision was required before it could be published on *Nat. Commun.*

1. A very similar work was published on Materials Today (<https://doi.org/10.1016/j.mattod.2022.02.014>) on 11 March, 2022, which used the same DLP technique and similar ink composed of HENPs, AAm, and PEGDA to fabricate 3D PC structures. What's the innovation or advantage of current work compared to that work?
2. In the introduction, the authors have emphasized the difficulties of 3D printing of PC structures multiple times, but the recent progresses in PC 3D printing were rarely introduced. With what kind of inks and printing process? What's the advantages or improvement of current work compared to the reported works?
3. The mechanism of colloid assembly in the DLP printing process was not explained clearly. On page 9, the authors mentioned "the continuous curing manner can provide a suction force for ink to continuously refill inward between the cured structure and the curing interface". The "suction force" has been mentioned throughout the paper. Where did the suction force come from? Should the motion of inks simply be attributed to the large surface tension of the liquid? Or a vacuum space formed during the lifting of the printed PC, which caused a pressure difference? Anyway, these physical phenomena may have nothing to do with the colloid assembly. It just determined the transportation of inks.
4. In Figure 3 and 4, it was not a scientific way to characterize the PC structure with normalized reflection peaks because it lost the intensity information, which are critical to evaluate the assembly quality and order degree of particle arrays.
5. How about the mechanical stability of the just printed 3D PC structure with such high-volume ratio of solvent? Will it affect the printing process, such as the maximum height of the printed object?
6. On page 11 and in Figure 3, the change of the reflection wavelength with the printing speed should be explained in the discussion.
7. On page 8, the author thinks that "...PS latex particles inside the polymer skeleton is uncontrollable without the suction force under discontinuous printing process...". Then, in the printing of segmental koi fish (Figure 4), did the printing process need to be paused to change the colloidal inks? What's the influence towards the PC structure?
8. The reflection peaks in Figure 4n was questionable. If the spectra were measured by a general optical fiber with collecting diameters of several millimeters, the reflection peak should be a doublet peak according to the size of the fish. If the spectra were measured by a microscale optical probe, the peak II, V, and VI actually reflected a transitional but still uniform lattice spacing between the neighboring PCs. Experimental proofs were required to support the reflection results.

Responses to Reviewer # 1

This manuscript reports continuous DLP printing of structurally colored 3D objects using colloidal suspensions. The suspensions are composed of PS@PMMA/PAA core-shell particles in the aqueous mixture of PEGDA, AAm, and a photoinitiator. Rich carboxylic groups on the surface of particles stabilize the suspension and make hydrogen bonds with AAm, which helps the particles to be evenly distributed in the final particle-polymer composite produced by photo-polymerization of AAm and PEGDA. 3D structures are continuously produced from a large drop of the suspension by the DLP technique during UV exposure, at which particles are embedded. After evaporation of water, particles are arranged to have some order, which develops structural colors in the shrunken 3D structures. As the drop of suspension is confined by capillary force, distinct suspensions can be consecutively employed to produce multi-colors in a series for single 3D structures. This work is generally interesting. But, the manuscript lacks detailed analysis. Furthermore, the authors might miss the recent paper on 3D printing of colloidal photonic structures, J. Liao et al. *Materials Today* (2022), <https://doi.org/10.1016/j.mattod.2022.02.014>.

Reply: We greatly appreciate the reviewer for the positive assessment and valuable suggestions. According to the reviewer's comments, the manuscript has been carefully revised. We hope that the revised manuscript would be suitable for publication in *Nature Communications*.

The authors should strengthen their work by addressing the following comments:

1. The 3D printing of structurally colored objects based on the DLP technique was reported recently in *Materials Today* (2022). The authors should modify the introduction and conclusion to review the previous work and justify the novelty and advances of this work.

Reply: Thanks for the reviewer's comments. We are sorry for missing the mentioned reference reported recently in *Materials Today* (2022). In the reference work, the concentrated highly charged elastic nanoparticles (HENPs) embedded in a UV-curable hydrogel system is mainly employed as the resin for 3D structuring. The resin itself can display corresponding structural color in the solution before 3D printing for structuring. **The coloration mechanism is the electrostatic repulsion among the charged nanoparticles according to their previous work (*Adv. Funct. Mater.* 2019, 39, 1902954), and is independent of the manufacturing method.** Due to the discontinuous printing mode, the fabrication efficiency is relatively low and the step structure along the printing direction is obvious (Resulting Structure row of **Table R1**), which reduces the printing fidelity, affects the structure continuity and the volumetric color intensity, where the structural color almost disappears when the viewing angle is 0° (Fig. 2g, h in the mentioned reference).

Our work employs **continuous DLP 3D printing** as the **coloration method**, and realizes the **simultaneous macroscopic printing and microscopic particle assembly** with hydrogen bonds assisted UV-curable structural color ink, which endows the 3D colloidal photonic crystal (PCs) structures with smooth sidewall (Resulting Structure row of **Table R1**) and brilliant **volumetric color property**. Due to the continuous printing mode, the printing fidelity and efficiency are remarkably improved.

Table R1. Comparison of the mentioned reference (*Mater. Today*, 2022) with this work concerning the preparation of 3D photonic crystal structure including preparation method, structural color generation mechanism and related-properties.

References		3D-printable colloidal photonic crystals, Mater. Today (2022) [1]	Our Work
Printing Process	3D Printing Method	Discontinuous digital light processing printing	Continuous digital light processing printing
	Fabrication Process	① Printing Process Build Platform Start Ink FEP Window UV Exposure Cured Structure UV Transition Black Screen UV Up-Movement UV Repeat ② Remove Structure	① Printing Process Continuous Spontaneous Assembly Suction Force UV ② Evaporation
	Structural Color Generation Mechanism	Assembly before printing	Assembly during printing
	Driving Force for Assembly	Electrostatic repulsive force between the highly charged elastic nanoparticles	Suction force induced by the continuous curing manner
	Structural Color Regulation Factors	Concentration of highly charged elastic nanoparticles, temperature and curing time	Particle diameter and printing speed
Structure Details and Properties	Resulting Structure	3D Layered geometry	3D Smooth geometry
	Step Eliminating Effect	Lateral layered structure	No steps; Smooth Surface
	Printing Fidelity	Limited fidelity due to the step structures and loss of structure continuity	High fidelity
	State of the Final Structure	Swollen in water	Rigid

	Volumetric color property	/	√
	Mechanical Property	 Enhanced mechanical strength	 Enhanced mechanical strength

Furthermore, the resulting 3D structure is in the dry state, and can be employed as optical device for practical use. Therefore, new light propagation experiments of the 3D printed PCs structures are conducted to prove the advantages of our method (**Figure R1**). Optical light-guide structures have attracted great attention for their ability to manipulate and transport light in a controlled manner, whose property relies on the medium morphology ^[2-6]. Based on the 3D printing induced coloration mechanism, hollow cylinder-shaped tubes with smooth inner and outer surfaces, low optical loss and color selectivity are continuously 3D printed from PS latex particles with different diameters (**Figure R1A-E**). In detail, when the white light is introduced into one opening of the structure, single-colored light corresponding to the photonic stopbands of PS latex is exported from the other opening (**Figure R1F**), exhibiting the frequency selective light-guide property. In addition, the optical loss is low and is basically linearly cylinder length related (**Figure R1G-L, Table R2**), where the optical loss coefficient is $\sim 4.57 \pm 1.05$ dB/cm and is comparable to the loss of light traveling the same distance in air ($\sim 3.28 \pm 0.14$ dB/cm). While for the hollow cylinder-shaped tube structures printed from the discontinuous printing method based on the same model (**Figure R2A-C**), they have poor surface smoothness and obvious step structures. Moreover, the printing stability decreases as the printing proceeds, leading to size deviation with narrow holes and limited length preparation, which results in the increased optical loss (the optical loss coefficient is $\sim 6.87 \pm 1.07$ dB/cm) and distorted output light pattern (**Figure R2D, E**). Thus, our 3D printing induced coloration mechanism and the feasibility in PCs structure printing is advantageous not only in fine 3D structure fabrication but also in optical device functionalization.

Figure R1 (Fig. 5 in the revised manuscript). Continuous DLP 3D printed PCs structures as color and pattern selective optical light-guide tubes. (A-E) Optical images of hollow cylinder-shaped tubes with different frequency selective light-guide property printed from PS particles with diameters of 192 nm (A), 214 nm (B), 230 nm (C), 245 nm (D) and 265 nm (E), respectively. (F) Schematic illustration of the length dependent

optical loss characterization. (G-L) Optical images of light-guide performance of hollow cylinder-shaped tubes with lengths of 9 mm (G, J), 18 mm (H, K) and 20 mm (I, L), respectively. J-L are the optical images of output light morphology on a black background from different lengths. (M-P) Optical images of output light patterns with triangle (M), square (N), pentagon (O) and flower (P) morphologies, respectively. The insets are cross-sectional morphology of corresponding tubes. (Q) Schematic illustration of light direction guidance through bended structures. (R-U) Optical images of bended optical light-guide structures with bending angle of 30° (R), 45° (S), 70° (T) and 90° (U), respectively. The insets are the corresponding models of optical light-guide structures. (V-X) Optical images of the light color regulation through overlapping lights guided from two light-guide tubes. The two light-guide tubes are printed from PS particles with diameters of 192 nm and 265 nm (V), 192 nm and 214 nm (W) and 265 nm and 214 nm (X), respectively.

Table R2 (Table S2 in the revised Supplementary Information). Optical losses of optical light-guide structures with different lengths.

Length (mm)	9	18	20
Optical Loss (dB)	5.37 ± 0.30	7.08 ± 0.17	7.65 ± 0.47

Figure R2 (Supplementary Figure 17 in the revised Supplementary Information). Optical transportation property of the discontinuous DLP 3D printed PCs structures. (A, B) Optical images of the discontinuously printed hollow cylinder tube structures with lengths of 9 mm (A) and 18 mm (B), respectively. (C) Enlarged optical image of the through-hole in the red dashed box in (B). (D, E) Optical images of the discontinuously printed hollow cylinder-shaped tubes with lengths of 9 mm (D) and 18 mm (E), respectively.

In addition, the output light pattern can also be manipulated based on the structural controllability. Triangular (Figure R1M), square (Figure R1N), pentagonal (Figure R1O) and flower-shaped (Figure R1P) colored light patterns can be output through the preparation of cylinder structures with different cross-section patterns. Besides the linear light guidance, it is also capable to change the output direction of light from 30° to 90° comparing with the input direction through continuous printing bended structures, as shown in Figure R1Q-U. Furthermore, interlapping multiple guided lights can generate new color of light with the rest non-overlapping areas retaining their original color (Figure R1V-X), which further confirms the morphology and color controllability of our method and its assurance in optical application.

Revisions in the manuscript:

(1) We have added the mentioned reference in the **Introduction** part with the description of “Though discontinuous 3D printing process³⁸ solves the problem in rapid preparation of 3D PCs structure, the rough surface and the poor fidelity still impede their applications in 3D optical devices. Therefore, the deterministic and large-scale fabrication of 3D structural color with smooth sidewall and brilliant volumetric color property through a simple and facile method remains a challenge.” in Line 2, Page 3 of the revised manuscript.

(2) Accordingly, the new light propagation experimental results of the continuous and discontinuous 3D printed PCs structures have been added as Fig. 5 in the revised manuscript, Table S2 and Supplementary Figure 17 in the revised Supporting Information, respectively, with the subheading as **Optical light-guides with color and pattern selectivity** and the description of: “Being capable of continuous 3D printing structures with high surface finish and volumetric color, the application in optical device preparation can thus be realized. Optical light-guide structures have attracted great attention for their ability to manipulate and transport light in a controlled manner, whose property relies on the medium morphology⁵⁶⁻⁶⁰. Based on the 3D printing induced coloration mechanism, hollow cylinder-shaped tubes with smooth inner and outer surfaces, low optical loss and color selectivity can be continuously 3D printed from PS latex particles with different diameters (Fig. 5a-e). In detail, when the white light is introduced into one opening of the structure, single-colored light corresponding to the photonic stopbands of PS latex is exported from the other opening (Fig. 5f), exhibiting the frequency selective light-guide property. In addition, the optical loss is low and is basically linearly cylinder length related (Fig. 5g-l, Table S2), where the optical loss coefficient is $\sim 4.57 \pm 1.05$ dB/cm and is comparable to the loss of light traveling the same distance in air ($\sim 3.28 \pm 0.14$ dB/cm). While for the hollow cylinder-shaped tube structures printed from the discontinuous printing method based on the same model (Supplementary Figure 17a-c), they have poor surface smoothness and obvious step structures. Moreover, the printing stability decreases as the printing proceeds, leading to size deviation with narrow holes and limited length preparation, which results in the increased optical loss (the optical loss coefficient is $\sim 6.87 \pm 1.07$ dB/cm) and distorted output light pattern (Supplementary Figure 17d, e).

In addition, the output light pattern can also be manipulated based on the structural controllability. Triangular (Fig. 5m), square (Fig. 5n), pentagonal (Fig. 5o) and flower-shaped (Fig. 5p) colored light patterns can be output through the preparation of cylinder structures with different cross-section patterns. Besides the linear light guidance, it is also capable to change the output direction of light from 30° to 90° comparing with the input direction through continuous printing bended structures, as shown in Fig. 5q-u. Furthermore, interlapping multiple guided lights can generate new color of light, and the rest non-overlapping areas retain their original color (Fig. 5v-x), which further confirms the morphology and color controllability of our method and its assurance in optical application. Thus, the printing induced coloration mechanism and the feasibility in PCs structure printing is advantageous not only in fine 3D structure fabrication but also in optical device functionalization.” in Line 20, Page 14 of the revised manuscript.

2. There is no discussion of how and when the particles produce ordered structures. First, the authors should analyze the internal structure of colloidal arrays to confirm whether the structures are crystalline or amorphous. Second, if the particles form crystalline arrays, the authors should study when the crystallization occurs and how the crystal orientation is determined (affected by the DLP process or drying?). It seems that the particles are not ordered in the printed structure before the evaporation. Is there any influence of the suction force and ink refilling on the crystallization process? If the structure is crystalline, it is strongly suggested to analyze the reflectance spectra with Bragg's equation.

Reply: Thanks for the reviewer's comments. We will answer the above comments point by point.

1) First, the authors should analyze the internal structure of colloidal arrays to confirm whether the structures are crystalline or amorphous.

Reply: After complete evaporation, the assembly of PS latex particles within the polymer skeleton is crystalline, as displayed in **Figure R3A**, the SEM image of the composite structure of PS-polymer skeleton

and **Figure R3B**, the SEM image of pure hydrogel after removing the PS latex particles, which displays hexagonal assembly.

2) Second, if the particles form crystalline arrays, the authors should study when the crystallization occurs and how the crystal orientation is determined (affected by the DLP process or drying?). It seems that the particles are not ordered in the printed structure before the evaporation. Is there any influence of the suction force and ink refilling on the crystallization process?

Reply: We have compared the surface and internal assembly of structures printed from the continuous 3D printing method (**Figure R3A**) and the directly curing method (**Figure R3C**), which are in fact the structures prepared from two methods both involve evaporation process, but with and without resin refilling process, respectively. The directly cured structure displays amorphous, while the continuous 3D printed structure displays crystalline, which means that bulk curing along with evaporation cannot realize crystalline assembly, while continuous layer-by-layer printing along with evaporation can lead to crystalline assembly. In addition, as the reviewer mentioned, the particles of the printed structure after printing and before evaporation don't seem ordered (**Figure R3D**), which can be ascribed to the existence of large amount of water induced large particle distance. Along with the amorphous assembly resulted from the discontinuous printing process (**Figure R3E**) without suction force, it can be concluded that **the suction force induced by the continuous DLP process and the continuous resin refilling determine the realization of crystalline assembly**, and the crystalline occurs after complete evaporation.

Figure R3. Assembly mechanism of the PS latex particles during the continuous DLP 3D printing process. (A, Fig. 2f in the revised manuscript) SEM images of the structure prepared by the continuous printing method after complete evaporation. (B, Fig. 2g in the revised manuscript) SEM images of the completely evaporated 3D structure after selectively removing the PS latex particles. (C, Supplementary Figure 2b, c in the revised Supplementary Information) SEM images of the directly cured structure. (D, Fig. 2d in the revised manuscript) SEM images of the just printed structure prepared by the continuous printing method. (E, Supplementary Figure 5c, d in the revised Supplementary Information)

3) If the structure is crystalline, it is strongly suggested to analyze the reflectance spectra with Bragg's equation.

Reply: According to the Bragg's equation, the stopband wavelength theoretically increases with the increasing of the PS particle diameter. As displayed in **Figure R4**, the measured results are consistent with the calculated ones, which further proves that the continuous 3D printed structure is crystalline.

Figure R4 (Supplementary Figure 11 in the revised Supplementary Information). The calculated and measured stopband wavelengths of the structures continuously 3D printed with different PS particle diameters.

Revisions in the manuscript:

(1) “How and when the particles produce ordered structures” have been revised with the description of “While for the same slab structure printed from discontinuous printing process (z-axis stepping length of 20 μm and staying time of 6 s for each layer), multiple separate steps are needed for printing one single layer⁵³ (Supplementary Figure 5a), where no low-pressure zone and suction force are generated, leading to uncontrollable assembly of PS latex particles inside the polymer skeleton and unobvious structural color (Supplementary Figure 5b-d). Along with the amorphous assembly of the directly cured structure (Supplementary Figure 2) which also excludes the suction force, it can be concluded that the suction force induced by the continuous DLP 3D printing and the continuous resin refilling determine the realization of crystalline assembly, and the crystalline occurs after complete evaporation.” in Line 10, Page 9 of the revised manuscript.

(2) Accordingly, we have added Figure R4 as Supplementary Figure 11 in the revised Supplementary Information with the description of “In addition, comparing with the assembly of pure PS latex particles, the interspersions of polymer skeletons among the PS latex particles increase the particle spacing, making the wavelength of the stopband red-shifted (Supplementary Figure 10), which are consistent with the calculated results (Supplementary Figure 11) by the Bragg’s law” in Line 7, Page 11 of the revised manuscript.

3. It is difficult to understand what the suction force is and how the suction force plays a role in particle ordering. Is there no suction force for stepwise printing?

Reply: Thanks for the reviewer’s comments. The suction force is originated from the low-pressure zone generated in the continuous printing process. Specifically, during the continuous printing process, the procedures of UV exposure, pattern change, cured structure separation and resin refilling are conducted simultaneously along with the continuous upward moving with the supporting plate. The cured structure is always immersed inside the uncured ink, creating a low-pressure zone under the previously printed layers, which induces the generation of suction force when the supporting plate is continuously moved up ^[7], as displayed in **Figure R5A**.

Figure R5. Comparison of printing process between the continuous and discontinuous DLP 3D printing. (A, **Fig. 2h in the revised manuscript**) Scheme of the continuous DLP 3D printing process. Suction force is generated from the low-pressure zone as the cured structure is always immersed inside the uncured ink. (B, **Supplementary Figure 5a in the revised Supplementary Information**) Scheme of the discontinuous DLP 3D printing process. As the resin refilling process is conducted separately and successively after upward lifting the supporting plate, no suction force occurs. (C, **Supplementary Figure 5c, d in the revised Supplementary Information**) SEM images of the structure printed by the discontinuous printing method. (D, **Fig. 2f in the revised manuscript**) SEM images of the structure prepared by the continuous printing method after complete evaporation.

While during the discontinuous printing process, multiple separate steps are needed for printing one single layer ^[8] (**Figure R5B**). In detail, the cured structure is firstly lifted upward for the separation of the cured structure from the curing interface, and then is pushed downward to the position for the next layer curing. Therefore, no pressure zone is generated under the discontinuous printing process. In addition, we have printed the same slab structure through discontinuous printing method without suction force. As displayed in **Figure R5C**, disordered assembly is acquired. Along with the continuous printing results with suction force in **Figure R5D**, it can be proved that the suction force plays a key role in particle ordering, which is in accordance with the reply of the 2nd comment of the reviewer: **the suction force induced by the continuous DLP process and the continuous resin refilling determine the realization of crystalline assembly and the crystalline occurs after complete evaporation.**

Revisions in the manuscript:

(1) To clearly illustrate the origin of suction force and its function in particle ordering during the continuous DLP 3D printing, accordingly, we have added Figure R5A as Fig. 2h as a new scheme with the description of “In detail, as displayed in Fig. 2h, during the continuous printing process, the cured structure is always

immersed inside the uncured ink, creating a low-pressure zone under the previously printed layers, which induces the generation of suction force when the supporting plate is continuously moved up⁵². The suction force can lead to the ink continuously refill inward between the cured structure and the curing interface, where the PS latex particles are solidified and confined assembled with a certain orientation inside the polymer skeleton with the assistance of hydrogen bonds (Fig. 2i).” in Line 21, Page 8, and the description of “Along with the amorphous assembly of the directly cured structure (Supplementary Figure 2) which also excludes the suction force, it can be concluded that the suction force induced by the continuous DLP 3D printing and the continuous resin refilling determine the realization of crystalline assembly, and the crystalline occurs after complete evaporation.” in Line 14, Page 9 of the revised manuscript.

(2) Accordingly, Figure R5B has been incorporated as Supplementary Figure 5a in the revised Supplementary Information with the description of “While for the same slab structure printed from discontinuous printing process (z-axis stepping length of 20 μm and staying time of 6 s for each layer), multiple separate steps are needed for printing one single layer⁵³ (Supplementary Figure 5a), where no low-pressure zone and suction force are generated, leading to uncontrollable assembly of PS latex particles inside the polymer skeleton and unobvious structural color (Supplementary Figure 5b-d).” in Line 10, Page 9 of the revised manuscript.

4. It is suggested to study why stopband position varies with printing speed.

Reply: Thanks for the reviewer’s suggestions. We have characterized the SEM images of the 3D PCs structures printed from 214 nm PS latex particles with different printing speeds. As displayed in **Figure R6**, the distance between particles increases with the increasing of printing speed, which leads to the red-shift of the stopband position for the same PS latex particle diameter.

Figure R6 (Supplementary Figure 14 in the revised Supplementary Information). Surface SEM images of the prepared PCs structure under different printing speeds.

As the different printing speeds require different UV intensities and exposure times of one layer, which leads to different printing times for the same 3D structure. For example, to print a pure AM hydrogel structure with design of 6.0 mm \times 6.0 mm \times 2.0 mm (z-axis height), the speed of 20 $\mu\text{m/s}$ needs 100.0 s under UV intensity of 5.7 mW/cm^2 , the speed of 40 $\mu\text{m/s}$ needs 50.0 s under UV intensity of 6.7 mW/cm^2 , the speed of 60 $\mu\text{m/s}$ needs 33.4 s under UV intensity of 7.2 mW/cm^2 , while the speed of 80 $\mu\text{m/s}$ requires 25.0 s under UV intensity of 7.6 mW/cm^2 . For different UV intensities and different printing times, the volumes before water evaporation are the same and correspond to designed volumes (**Figure R7A**), while the dry volume after water evaporation for the sample cured fast under higher UV intensity is larger than that of lower UV intensity (**Figure R7B**). The results indicate that the higher printing speed under higher UV intensity will lead to larger dry volume of polymer skeleton and a smaller ratio of PS latex particles solidified inside the skeleton, resulting in a larger latex particle spacing and the red-shift of the stopband position, which is in consistence with the stopband wavelength characterization and particle spacing results.

Figure R7 (Supplementary Figure 13 in the revised Supplementary Information). Volumes of the pure AM hydrogels prepared under different printing speeds before (A) and after (B) complete evaporation.

Revisions in the manuscript: The new experimental results have been added as Supplementary Figure 13 and Supplementary Figure 14 in the revised Supplementary Information with the description of “As different printing speeds require different UV intensities and exposure times of one layer, which leads to different printing times and different ratios of PS particles for the same 3D structure. In detail, higher printing speed under higher UV intensity will lead to larger dry volume of polymer skeleton (Supplementary Figure 13) and a smaller ratio of PS latex particles solidified inside the skeleton, resulting in a larger particle spacing (Supplementary Figure 14) and a red-shifted structural color.” in Line 5, Page 12 of the revised manuscript.

5. How do the PS@PMMA/PAA core-shell particles maintain high suspension stability in the aqueous mixture of PEGDA, AAm, and photoinitiator despite the possible charge screening by AAm? It would be helpful to measure the zeta potentials of the particles in distilled water and the aqueous mixture.

Reply: Thanks for the reviewer’s comments. According to the reviewer’s suggestion, we have measured the zeta potentials of PS@PMMA/PAA core-shell latex particles (PS latex particles) in the pure aqueous solution and the aqueous mixture by the Zetasizer Nano (Nano ZS ZEN3600, Malvern Instrument, UK), which are -36.07 ± 0.75 mV and -30.77 ± 0.21 mV, respectively. Thus, the adding of UV-curable system does not influence the stability of particles inside the system.

Revisions in the manuscript: We have added the zeta potential values with the description of “In addition, zeta potential of the UV-curable structural color ink (-30.77 ± 0.21 mV) does not change significantly comparing to the PS latex particles in the pure aqueous solution (-36.07 ± 0.75 mV), which ensures a high suspension stability for 3D printing.” in Line 8, Page 6, and the description of “Zeta potentials of PS latex particles in the pure aqueous solution and the aqueous mixture are measured by Zetasizer Nano (Nano ZS ZEN3600, Malvern Instrument, UK). Each reported zeta potential is an average of at least three independent measurements.” has been added in the “**Characterization**” part of **Methods** section of the revised manuscript.

6. The boundary between two adjacent parts with different colors is sharply defined in Figure 4? What causes a single reflectance peak, instead of two peaks, at the boundary which is positioned between the peaks of the two adjacent parts?

Reply: Thanks for the reviewer’s comments. The reason of a single reflectance peak at the boundary, instead of two peaks, is ascribed to the detection method of the reflectance spectra, which displays a composite color rather than two separate colors. As the interface is sharp and is on the microscale, we measure the reflectance spectra by the fibre optic spectrometer (NoVA-EX, IdeaOptics, China) through an optical microscope (BX53, OLYMPUS, Japan). The detection diameter is related to the magnification of objective lens, which is usually

tens of micrometers. In our experiment, the reflectance spectra of interface between the parts printed from particle diameters of 192 nm and 214 nm are measured at magnification of 50X, whose detection diameter is $\sim 10 \mu\text{m}$. In addition, we also measure the same interface from the lens of 20X with detection diameter of $\sim 25 \mu\text{m}$. As shown in **Figure R8**, the reflectance spectrum also displays one stopband, whose position is the same as the one characterized from 50X lens, confirming that the detection range of optical lens in the microscale displays the composite color rather than two single colors.

Figure R8. The reflection spectra measurement under different magnifications of the interface between two adjacent parts.

Revisions in the manuscript: We have added the reflectance characterization method with the description of “Reflectance spectra are measured by fibre optic spectrometer (NoVA-EX, IdeaOptics, China) through an optical microscope (BX53, OLYMPUS, Japan).” in the “**Characterization**” part of **Method** section of the revised manuscript.

7. What causes the broadening of reflectance peak for 3D printed structure in Supplementary Figure 10? It is suggested to the absolute unit for all reflectance spectra. What is the gravitational deposition method?

Reply: Thanks for the reviewer’s comments. We have replied in the 2nd comment of the reviewer, the suction force determines the crystalline assembly of the continuous 3D printed PCs structure. As the printing time is larger than $20 \mu\text{m/s}$, which is much faster than the assembly time of the pure PS latex particles under constant temperature and humidity through the gravitational deposition method. It is much easier to generate defects under such high printing speed, which may lead to the broadening of the reflectance peak. In addition, the introduction of hydrogel interspersing the gaps among the PS latex particles may also lead to the broadening of the reflectance peak than the pure latex particles assembly ^[9-13], which is a common phenomenon for the composite nanoparticle-hydrogel structures.

For the unit of the reflectance spectra, current PCs 3D printing methods cited in the introduction part employ the normalized reflection peaks to characterize the PC structure, which is also commonly used in other methods for PCs structures preparation ^[9, 11, 14]. In addition, as the absolute value is related with the test equipment and test conditions (such as light energy and luminous flux), normalized intensity is more commonly employed for comparison of different references. Thus, the normalized intensity, rather than absolute intensity, is utilized throughout the manuscript.

The gravitational deposition is a traditional method to assemble latex particles ^[15-18] and the scheme is shown in **Figure R9**, where the monodisperse latex particles continuously move downward in the dispersion under

the force of gravity in constant temperature and humidity, and finally self-assemble into ordered photonic crystal structure after the complete solvent evaporation.

Figure R9 (Supplementary Figure 10a in the revised Supplementary Information). Scheme of the gravitational deposition method.

Revisions in the manuscript:

(1) Accordingly, the schematic diagram of gravitational deposition method has been added as Supplementary Figure 10a in the revised manuscript.

(2) The reasons that lead to the reflectance peak of 3D printed structure broaden have been added with the description of “Comparing with the pure PS latex particles assembled by gravitational deposition method, the broadening of reflectance peak of the 3D printed structure can be ascribed to the high printing speed and assembly speed. In addition, the introduction of hydrogel interspersing the gaps among the PS latex particles may also lead to the broadening of the reflectance peak than the pure latex particles assembly^[5-9], which is a common phenomenon for the composite nanoparticle-hydrogel structures.” in the revised Supplementary Information.

8. What is the slicing thickness on page 10?

Reply: Thanks for the reviewer’s comments. We are sorry for missing the slicing thickness on Page 10, which is 5.0 μm . Except for the investigation of the influence of slicing thickness on the stopband wavelength in Supplementary Figure 8, the slicing thickness for 3D printing is set as 5.0 μm .

Revisions in the manuscript: We have supplemented the sentences “The light pattern sequences played by the UV projector are acquired through slicing the 3D model into a series of cross-sectional images with slicing thicknesses of 5 μm . To investigate the influence of slicing thickness on the stopband wavelength, slicing thickness of 20 μm and 50 μm are also employed.” in the “**3D printing apparatus**” part of **Methods** section of the revised manuscript.

References

1. Liao, J. *et al.* 3D-printable colloidal photonic crystals. *Mater. Today* **56**, 29-41 (2022).
2. Parker, S. T. *et al.* Biocompatible Silk Printed Optical Waveguides. *Adv. Mater.* **21**, 2411-2415 (2009).
3. Lorang, D. J. *et al.* Photocurable Liquid Core-Fugitive Shell Printing of Optical Waveguides. *Adv. Mater.* **23**, 5055-5058 (2011).
4. Frascella, F. *et al.* Three-Dimensional Printed Photoluminescent Polymeric Waveguides. *ACS Appl. Mater. Interfaces* **10**, 39319-39326 (2018).
5. Nizamoglu, S. *et al.* Bioabsorbable polymer optical waveguides for deep-tissue photomedicine. *Nat. Commun.* **7**, 10374 (2016).
6. Choi, M. *et al.* Light-guiding hydrogels for cell-based sensing and optogenetic synthesis in vivo. *Nat. Photonics* **7**, 987-994 (2013).

7. Wang, K. *et al.* 3D Printing of Viscoelastic Suspensions via Digital Light Synthesis for Tough Nanoparticle–Elastomer Composites. *Adv. Mater.* **32**, 2001646 (2020).
8. Tumbleston, J. R. *et al.* Continuous liquid interface production of 3D objects. *Science* **347**, 1349-1352 (2015).
9. Zhang, Y. *et al.* Super-Elastic Magnetic Structural Color Hydrogels. *Small* **15**, e1902198 (2019).
10. Takeoka, Y. *et al.* Production of colored pigments with amorphous arrays of black and white colloidal particles. *Angew. Chem. Int. Ed.* **52**, 7261-7265 (2013).
11. Fu, F., Shang, L., Chen, Z., Yu, Y. & Zhao, Y. Bioinspired living structural color hydrogels. *Sci. Robot.* **3**, eaar8580 (2018).
12. Kim, H. *et al.* Structural colour printing using a magnetically tunable and lithographically fixable photonic crystal. *Nat. Photonics* **9**, 534-540 (2009).
13. Ohtsuka, Y., Seki, T. & Takeoka, Y. Thermally Tunable Hydrogels Displaying Angle-Independent Structural Colors. *Chem. Int. Ed.* **54**, 15368-15373 (2015).
14. Zhao, Z. *et al.* Bioinspired Heterogeneous Structural Color Stripes from Capillaries. *Adv. Mater.* **29**, 1704569 (2017).
15. Jiang, P., Bertone, J. F., Hwang, K. S. & Colvin, V. L. Single-crystal colloidal multilayers of controlled thickness. *Chem. Mater.* **11**, 2132-2140 (1999).
16. Huang, Y. *et al.* Colloidal photonic crystals with narrow stopbands assembled from low-adhesive superhydrophobic substrates. *J. Am. Chem. Soc.* **134**, 17053-17058 (2012).
17. Wang, J., Wen, Y., Feng, X., Song, Y. & Jiang, L. Control over the Wettability of Colloidal Crystal Films by Assembly Temperature. *Macromol. Rapid Commun.* **27**, 188-192 (2006).
18. von Freymann, G., Kitaev, V., Lotsch, B. V. & Ozin, G. A. Bottom-up assembly of photonic crystals. *Chem. Soc. Rev.* **42**, 2528-2554 (2013).

Responses to Reviewer # 2

In this manuscript, 3D photonic crystal with volumetric color was prepared by a continuous digital light processing (DLP) 3D printing method. The preparation of colloid ink and the control of the printing process was demonstrated. DLP technique with an ink composed of acrylamide and PEGDA have been widely studied to fabricate 3D biomaterials (Adv. Healthcare. Mater. 2020, 9, 2000156), and it is a good idea to extend this technique to the fabrication of 3D PC structures. However, some statements in this paper could not be well supported/explained by the discussion. Therefore, a major revision was required before it could be published on Nat. Commun.

Reply: We thank the reviewer very much for the positive comments and the suggestions for improving the manuscript. According to the reviewer's comments, we have made revisions on our manuscript and addressed them one by one as below. We hope that the revised manuscript would be suitable for publication in *Nature Communications*.

1. A very similar work was published on Materials Today (<https://doi.org/10.1016/j.mattod.2022.02.014>) on 11 March, 2022, which used the same DLP technique and similar ink composed of HENPs, AAm, and PEGDA to fabricate 3D PC structures. What's the innovation or advantage of current work compared to that work?

Reply: Thanks for the reviewer's comments. We are sorry for missing the mentioned reference reported recently in Materials Today (2022). In the reference work, the concentrated highly charged elastic nanoparticles (HENPs) embedded in a UV-curable hydrogel system is mainly employed as the resin for 3D structuring. The resin itself can display corresponding structural color in the solution before 3D printing for structuring. **The coloration mechanism is the electrostatic repulsion among the charged nanoparticles according to their previous work (Adv. Funct. Mater. 2019, 39, 1902954), and is independent of the manufacturing method.** Due to the discontinuous printing mode, the fabrication efficiency is relatively low and the step structure along the printing direction is obvious (Resulting Structure row of **Table R1**), which reduces the printing fidelity, affects the structure continuity and the volumetric color intensity, where the structural color almost disappears when the viewing angle is 0° (Fig. 2g, h in the mentioned reference).

Our work employs **continuous DLP 3D printing** as the **coloration method**, and realizes the **simultaneous macroscopic printing and microscopic particle assembly** with hydrogen bonds assisted UV-curable structural color ink, which endows the 3D colloidal photonic crystal (PCs) structures with smooth sidewall (Resulting Structure row of **Table R1**) and brilliant **volumetric color property**. Due to the continuous printing mode, the printing fidelity and efficiency are remarkably improved.

Table R1. Comparison of the mentioned reference (Mater. Today, 2022) with this work concerning the preparation of 3D photonic crystal structure including preparation method, structural color generation mechanism and related-properties.

References	3D-printable colloidal photonic crystals, Mater. Today (2022) ^[1]	Our Work
------------	--	----------

Printing Process	3D Printing Method	Discontinuous digital light processing printing	Continuous digital light processing printing
	Fabrication Process		
	Structural Color Generation Mechanism	Assembly before printing	Assembly during printing
	Driving Force for Assembly	Electrostatic repulsive force between the highly charged elastic nanoparticles	Suction force induced by the continuous curing manner
	Structural Color Regulation Factors	Concentration of highly charged elastic nanoparticles, temperature and curing time	Particle diameter and printing speed

Structure Details and Properties	Resulting Structure	 3D Layered geometry	 3D Smooth geometry
	Step Eliminating Effect	Lateral layered structure	No steps; Smooth Surface
	Printing Fidelity	Limited fidelity due to the step structures and loss of structure continuity	High fidelity
	State of the Final Structure	Swollen in water	Rigid
	Volumetric color property	/	√
	Mechanical Property	 Enhanced mechanical strength	 Enhanced mechanical strength

Furthermore, the resulting 3D structure is in the dry state, and can be employed as optical device for practical use. Therefore, new light propagation experiments of the 3D printed PCs structures are conducted to prove the advantages of our method (**Figure R1**). Optical light-guide structures have attracted great attention for their ability to manipulate and transport light in a controlled manner, whose property relies on the medium morphology ^[2-6]. Based on the 3D printing induced coloration mechanism, hollow cylinder-shaped tubes with smooth inner and outer surfaces, low optical loss and color selectivity are continuously 3D printed from PS latex particles with different diameters (**Figure R1A-E**). In detail, when the white light is introduced into one opening of the structure, single-colored light corresponding to the photonic stopbands of PS latex is exported from the other opening (**Figure R1F**), exhibiting the frequency selective light-guide property. In addition, the optical loss is low and is basically linearly cylinder length related (**Figure R1G-L, Table R2**), where the optical loss coefficient is $\sim 4.57 \pm 1.05$ dB/cm and is comparable to the loss of light traveling the same distance in air ($\sim 3.28 \pm 0.14$ dB/cm). While for the hollow cylinder-shaped tube structures printed from the discontinuous printing method based on the same model (**Figure R2A-C**), they have poor surface smoothness and obvious step structures. Moreover, the printing stability decreases as the printing proceeds, leading to size deviation with narrow holes and limited length preparation, which results in the increased optical loss (the optical loss coefficient is $\sim 6.87 \pm 1.07$ dB/cm) and distorted output light pattern (**Figure R2D, E**). Thus, our 3D printing induced coloration mechanism and the feasibility in PCs structure printing is advantageous not only in fine 3D structure fabrication but also in optical device functionalization.

Figure R1 (Fig. 5 in the revised manuscript). Continuous DLP 3D printed PCs structures as color and pattern selective optical light-guide tubes. (A-E) Optical images of hollow cylinder-shaped tubes with different frequency selective light-guide property printed from PS particles with diameters of 192 nm (A), 214 nm (B), 230 nm (C), 245 nm (D) and 265 nm (E), respectively. (F) Schematic illustration of the length dependent optical loss characterization. (G-L) Optical images of light-guide performance of hollow cylinder-shaped tubes with lengths of 9 mm (G, J), 18 mm (H, K) and 20 mm (I, L), respectively. J-L are the optical images of output light morphology on a black background from different lengths. (M-P) Optical images of output light patterns with triangle (M), square (N), pentagon (O) and flower (P) morphologies, respectively. The insets are cross-sectional morphology of corresponding tubes. (Q) Schematic illustration of light direction guidance through bended structures. (R-U) Optical images of bended optical light-guide structures with bending angle of 30° (R), 45° (S), 70° (T) and 90° (U), respectively. The insets are the corresponding models of optical light-guide structures. (V-X) Optical images of the light color regulation through overlapping lights guided from two light-guide tubes. The two light-guide tubes are printed from PS particles with diameters of 192 nm and 265 nm (V), 192 nm and 214 nm (W) and 265 nm and 214 nm (X), respectively.

Table R2 (Table S2 in the revised Supplementary Information). Optical losses of optical light-guide structures with different lengths.

Length (mm)	9	18	20
Optical Loss (dB)	5.37 ± 0.30	7.08 ± 0.17	7.65 ± 0.47

Figure R2 (Supplementary Figure 17 in the revised Supplementary Information). Optical transportation property of the discontinuous DLP 3D printed PCs structures. (A, B) Optical images of the discontinuously

printed hollow cylinder tube structures with lengths of 9 mm (A) and 18 mm (B), respectively. (C) Enlarged optical image of the through-hole in the red dashed box in (B). (D, E) Optical images of the discontinuously printed hollow cylinder-shaped tubes with lengths of 9 mm (D) and 18 mm (E), respectively.

In addition, the output light pattern can also be manipulated based on the structural controllability. Triangular (Figure R1M), square (Figure R1N), pentagonal (Figure R1O) and flower-shaped (Figure R1P) colored light patterns can be output through the preparation of cylinder structures with different cross-section patterns. Besides the linear light guidance, it is also capable to change the output direction of light from 30° to 90° comparing with the input direction through continuous printing bended structures, as shown in Figure R1Q-U. Furthermore, interlapping multiple guided lights can generate new color of light with the rest non-overlapping areas retaining their original color (Figure R1V-X), which further confirms the morphology and color controllability of our method and its assurance in optical application.

Revisions in the manuscript:

(1) We have added the mentioned reference in the **Introduction** part with the description of “Though discontinuous 3D printing process³⁸ solves the problem in rapid preparation of 3D PCs structure, the rough surface and the poor fidelity still impede their applications in 3D optical devices. Therefore, the deterministic and large-scale fabrication of 3D structural color with smooth sidewall and brilliant volumetric color property through a simple and facile method remains a challenge.” in Line 2, Page 3 of the revised manuscript.

(2) Accordingly, the new light propagation experimental results of the continuous and discontinuous 3D printed PCs structures have been added as Fig. 5 in the revised manuscript, Table S2 and Supplementary Figure 17 in the revised Supporting Information, respectively, with the subheading as **Optical light-guides with color and pattern selectivity** and the description of: “Being capable of continuous 3D printing structures with high surface finish and volumetric color, the application in optical device preparation can thus be realized. Optical light-guide structures have attracted great attention for their ability to manipulate and transport light in a controlled manner, whose property relies on the medium morphology⁵⁶⁻⁶⁰. Based on the 3D printing induced coloration mechanism, hollow cylinder-shaped tubes with smooth inner and outer surfaces, low optical loss and color selectivity can be continuously 3D printed from PS latex particles with different diameters (Fig. 5a-e). In detail, when the white light is introduced into one opening of the structure, single-colored light corresponding to the photonic stopbands of PS latex is exported from the other opening (Fig. 5f), exhibiting the frequency selective light-guide property. In addition, the optical loss is low and is basically linearly cylinder length related (Fig. 5g-l, Table S2), where the optical loss coefficient is $\sim 4.57 \pm 1.05$ dB/cm and is comparable to the loss of light traveling the same distance in air ($\sim 3.28 \pm 0.14$ dB/cm). While for the hollow cylinder-shaped tube structures printed from the discontinuous printing method based on the same model (Supplementary Figure 17a-c), they have poor surface smoothness and obvious step structures. Moreover, the printing stability decreases as the printing proceeds, leading to size deviation with narrow holes and limited length preparation, which results in the increased optical loss (the optical loss coefficient is $\sim 6.87 \pm 1.07$ dB/cm) and distorted output light pattern (Supplementary Figure 17d, e).

In addition, the output light pattern can also be manipulated based on the structural controllability. Triangular (Fig. 5m), square (Fig. 5n), pentagonal (Fig. 5o) and flower-shaped (Fig. 5p) colored light patterns can be output through the preparation of cylinder structures with different cross-section patterns. Besides the linear light guidance, it is also capable to change the output direction of light from 30° to 90° comparing with the input direction through continuous printing bended structures, as shown in Fig. 5q-u. Furthermore, interlapping multiple guided lights can generate new color of light, and the rest non-overlapping areas retain their original color (Fig. 5v-x), which further confirms the morphology and color controllability of our method and its assurance in optical application. Thus, the printing induced coloration mechanism and the feasibility

in PCs structure printing is advantageous not only in fine 3D structure fabrication but also in optical device functionalization.” in Line 20, Page 14 of the revised manuscript.

2. In the introduction, the authors have emphasized the difficulties of 3D printing of PC structures multiple times, but the recent progresses in PC 3D printing were rarely introduced. With what kind of inks and printing process? What’s the advantages or improvement of current work compared to the reported works?

Reply: Thanks for the reviewer’s comments. Accordingly, we have concluded the current methods of PCs 3D printing mentioned in the introduction part including inkjet printing, direct ink writing, fused deposition modeling and two-photon polymerization lithography, as well as the discontinuous DLP 3D printing mentioned by the reviewer, based on the building blocks of colloidal particles, liquid crystals, block copolymers and resins. The printing process, structural color ink, structure details as well as properties of the printed 3D structure are summarized as the supporting information to prove the advantage of this work (**Table R3**). In detail, inkjet printing and direct ink writing are limited by the low construction freedom in 3D and are constrained to the discrete and plane shapes. Two-photon polymerization lithography are limited by the print dimension and productivity, while fused deposition modeling and discontinuous 3D printing are limited by the surface smoothness and the weak volumetric structural color, which impede their applications in 3D optical devices.

Our work can solve the above-mentioned limitations, and can realize arbitrary, deterministic and large-scale fabrication of smooth 3D structural color with volumetric color property through the **simultaneous macroscopic printing and microscopic particle assembly** mechanism endowed by the continuous DLP 3D printing method, and extend the application to the construction of customized jewelry accessories, decoration and optical device functionalization.

Table R3 (Table S1 in the revised Supplementary Information). Comparison of researches with this work concerning the preparation of 3D photonic crystal structure including preparation method, structural color generation mechanism and related-properties.

References		Facile full-color printing with a single transparent ink, Sci. Adv. , 2021 ^[7]	Tunable structural color of bottlebrush block copolymers through direct-write 3D printing from solution, Sci. Adv. , 2020 ^[8]	Structural color for additive manufacturing: 3D-printed photonic crystals from block copolymers, ACS Nano , 2017 ^[9]	Structural color three-dimensional printing by shrinking photonic crystals, Nat. Commun. , 2019 ^[10]	3D-printable colloidal photonic crystals, Mater. Today , 2022 ^[1]	Our Work
Fabrication Process	3D Printing Method	Inkjet printing	Direct ink writing	Fused deposition modeling	Two-photon polymerization lithography	Discontinuous digital light processing printing	Continuous digital light processing printing
	Printing Process							Structural Color Generation Mechanism	Assembly during printing	Assembly before printing	Assembly during printing	Coloration after heat-induced shrinking	Assembly before printing	Assembly during printing

	Driving Force for Assembly	Surface tension	Capillary flow during drying	Thermally induced during filament extrusion	/	Electrostatic repulsive force between the highly charged elastic nanoparticles	Suction force induced by the continuous curing manner
	Structural Color Regulation Factors	Ink volume and substrate wettability	Printing speed and substrate temperature	Molecular weight of block copolymer	Lattice constants and laser power	Concentration of highly charged elastic nanoparticles, temperature and curing time	Particle diameter and printing speed
Structural Color Ink	3D Printable Ink	Transparent UV-curable polymer ink	Bottlebrush block copolymer solution	Dendritic block copolymers	Commercial acrylate-based photoresist	Acrylamide-based UV-curable ink with embedded HENPs	Acrylamide-based UV-curable ink with dispersed carboxyl-rich PS latex particles
	Structural Color Generation Principle	Total internal reflection from micro-dome	Reflection	Reflection	Bragg scattering	Bragg scattering	Bragg scattering
Structure Details and Properties	Resulting Structure	Plane shape 	Plane shape 	3D geometry 	3D geometry 	3D geometry 	3D geometry 	Step Eliminating Effect	/	/	Lateral stripes and Rough surface	/	Lateral layered structure;	No steps; Smooth Surface
	Printing Fidelity	High fidelity	/	/	/	Limited fidelity due to the step structures and loss of structure continuity	High fidelity
	State of the Final Structure	Rigid	Rigid	Rigid	Rigid	Flexible and swollen equilibrium in water	Rigid
	Volumetric color property	/	/	/	✓	/	✓
	Mechanical Property	/	/	/	/	 Enhanced mechanical strength	 Enhanced mechanical strength

Revisions in the manuscript: Accordingly, we have added Table R3 as Table S1 in the revised Supplementary Information with the description of “3D printing can fabricate arbitrary geometries without the template prefabrication, etching or masking required in the traditional process^{22, 23}, and have been employed to construct complex 3D photonic structures. (Table S1). In detail, inkjet printing²⁴⁻²⁶, direct ink writing²⁷⁻²⁹ and fused deposition modeling³⁰ have been demonstrated to prepare patterned structural color from various building blocks including colloidal particles^{31, 32}, liquid crystals^{33, 34} or block copolymers^{35, 36}. However, the construction freedom in 3D is low and limited to discrete and plane shapes. Moreover, the cumbersome equilibrium coloration process and the weak volumetric structural color impede their wide applications. Two-photon polymerization printing of lattice structures³⁷ has been proved to realize the 3D color fabrication, but is compromised markedly by the print dimension and productivity. Though discontinuous 3D printing process³⁸ solves the problem in rapid preparation of 3D PCs structure, the rough surface and the poor fidelity still impede their applications in 3D optical devices. Therefore, the deterministic and large-scale fabrication of 3D structural color with smooth sidewall and brilliant volumetric color property through a simple and facile method remains a challenge.” in Line 14, Page 2 of the revised manuscript.

3. The mechanism of colloid assembly in the DLP printing process was not explained clearly. On page 9, the authors mentioned “the continuous curing manner can provide a suction force for ink to continuously refill inward between the cured structure and the curing interface”. The “suction force” has been mentioned throughout the paper. Where did the suction force come from? Should the motion of inks simply be attributed to the large surface tension of the liquid? Or a vacuum space formed during

the lifting of the printed PC, which caused a pressure difference? Anyway, these physical phenomena may have nothing to do with the colloid assembly. It just determined the transportation of inks.

Reply: Thanks for the reviewer’s comments. As the reviewer mentioned, the suction force is originated from the vacuum space and the low-pressure zone generated in the continuous printing process ^[11], as displayed in **Figure R3A**.

Figure R3. Assembly mechanism of the PS latex particles during the continuous DLP 3D printing. (**A, Fig. 2h in the revised manuscript**) Scheme of the continuous DLP 3D printing process. Suction force is generated from the vacuum space and the low-pressure zone. (**B, Fig. 2f in the revised manuscript**) SEM images of the structure prepared by the continuous printing method after complete evaporation. (**C, Supplementary Figure 2b, c in the revised Supplementary Information**) SEM images of the directly cured structure. (**D, Supplementary Figure 5a in the revised Supplementary Information**) Scheme of the discontinuous DLP 3D printing process. The resin refilling process is conducted independently after upward lifting the supporting plate, where no suction force occurs. (**E, Supplementary Figure 5c, d in the revised Supplementary Information**) SEM images of the structure printed by the discontinuous printing method.

For the function of suction force, we have compared the continuous 3D printed structure (**Figure R3B**) and the directly cured structure (**Figure R3C**), it can be found that the directly cured structure is amorphous, while the continuous 3D printed structure is well assembled. The fabrication process of both involves the evaporation step, but the difference is with or without the ink refilling induced by the continuous 3D printing. While for the discontinuous 3D printing, no pressure zone is generated as the cured structure is firstly lifted upward for the separation of the cured structure from the curing interface, and then is pushed downward to the position for the next layer curing ^[12] (**Figure R3D**). The particle assembly of the same slab structure printed from discontinuous printing method is disordered (**Figure R3E**). Therefore, by comparing with the directly cured structure and the discontinuous printed structure, it can be concluded that the suction force induced by the continuous DLP 3D printing determines the realization of crystalline assembly.

Revisions in the manuscript:

(1) To clearly illustrate the origin of suction force and its function in particle ordering during the continuous DLP 3D printing, accordingly, we have added Figure R3A as Fig. 2h as a new scheme with the description of “In detail, as displayed in Fig. 2h, during the continuous printing process, the cured structure is always immersed inside the uncured ink, creating a low-pressure zone under the previously printed layers, which induces the generation of suction force when the supporting plate is continuously moved up⁵². The suction force can lead to the ink continuously refill inward between the cured structure and the curing interface, where the PS latex particles are solidified and confined assembled with a certain orientation inside the polymer skeleton with the assistance of hydrogen bonds (Fig. 2i).” in Line 21, Page 8, and the description of “Along with the amorphous assembly of the directly cured structure (Supplementary Figure 2) which also excludes the suction force, it can be concluded that the suction force induced by the continuous DLP 3D printing and the continuous resin refilling determine the realization of crystalline assembly, and the crystalline occurs after complete evaporation.” in Line 14, Page 9 of the revised manuscript.

(2) Accordingly, Figure R3D has been incorporated as Supplementary Figure 5a in the revised Supplementary Information with the description of “While for the same slab structure printed from discontinuous printing process (z-axis stepping length of 20 μm and staying time of 6 s for each layer), multiple separate steps are needed for printing one single layer⁵³ (Supplementary Figure 5a), where no low-pressure zone and suction force are generated, leading to uncontrollable assembly of PS latex particles inside the polymer skeleton and unobvious structural color (Supplementary Figure 5b-d).” in Line 10, Page 9 of the revised manuscript.

4. In Figure 3 and 4, it was not a scientific way to characterize the PC structure with normalized reflection peaks because it lost the intensity information, which are critical to evaluate the assembly quality and order degree of particle arrays.

Reply: Thanks for the reviewer’s suggestions. We have summarized the current PCs 3D printing methods in replying the 1st and 2nd comments of the reviewer, where most of the references employ the normalized reflection peaks to characterize the PC structure. In addition, the normalized reflection peak is also commonly used in other methods for preparing PCs structures ^[13-15]. Moreover, as the absolute value is related with the test equipment and test conditions (such as light energy and luminous flux), normalized intensity is more commonly employed for comparison of different references. Therefore, the normalized reflection peak, rather than absolute intensity, is utilized throughout the manuscript.

5. How about the mechanical stability of the just printed 3D PC structure with such high-volume ratio of solvent? Will it affect the printing process, such as the maximum height of the printed object?

Reply: Thanks for the reviewer’s comments. According to the reviewer’s suggestion, we have conducted the tensile stress-strain measurement of the just printed 3D PCs structure to investigate the mechanical stability. During the continuous 3D printing process, the adhesion between the printed 3D PCs structure and the curing interface may lead to the limited printing height and decrease the stability.

Figure R4. Comparison of the real-time force-displacement curves of the just printed 3D PCs structure and the separation force between the just cured structure and the curing interface.

In our experiment, the low-adhesive curing interface is employed, which has been proved that the damage to both the curing interface and the cured structure is little during the continuous printing process ^[16,17] due to the low separation force and the short separation distance. As shown in **Figure R4**, the just printed 3D PCs structure ruptures at 1.45 N with separation distance of 72.95 mm, which are much larger than the adhesion induced by the continuous curing (separation force of tens of millinewtons and separation distance of tens of micrometers), indicating that the just printed 3D PCs structure is mechanically stable to ensure the continuous printing method.

6. On page 11 and in Figure 3, the change of the reflection wavelength with the printing speed should be explained in the discussion.

Reply: Thanks for the reviewer’s suggestions. We have characterized the SEM images of the 3D PCs structures printed from 214 nm PS latex particles with different printing speeds. As displayed in **Figure R5**, the distance between particles increases with the increasing of printing speed, which leads to the red-shift of the stopband position for the same PS latex particle diameter.

Figure R5 (Supplementary Figure 14 in the revised Supplementary Information). Surface SEM images of the prepared PCs structure under different printing speeds.

As the different printing speeds require different UV intensities and exposure times of one layer, which leads to different printing times for the same 3D structure. For example, to print a pure AM hydrogel structure with design of 6.0 mm × 6.0 mm × 2.0 mm (z-axis height), the speed of 20 µm/s needs 100.0 s under UV intensity

of 5.7 mW/cm², the speed of 40 μm/s needs 50.0 s under UV intensity of 6.7 mW/cm², the speed of 60 μm/s needs 33.4 s under UV intensity of 7.2 mW/cm², while the speed of 80 μm/s requires 25.0 s under UV intensity of 7.6 mW/cm². For different UV intensities and different printing times, the volumes before water evaporation are the same and correspond to designed volumes (**Figure R6A**), while the dry volume after water evaporation for the sample cured fast under higher UV intensity is larger than that of lower UV intensity (**Figure R6B**). The results indicate that the higher printing speed under higher UV intensity will lead to larger dry volume of polymer skeleton and a smaller ratio of PS latex particles solidified inside the skeleton, resulting in a larger latex particle spacing and the red-shift of the stopband position, which is in consistence with the stopband wavelength characterization and particle spacing results.

Figure R6 (Supplementary Figure 13 in the revised Supplementary Information). Volumes of the pure AM hydrogels prepared under different printing speeds before (A) and after (B) complete evaporation.

Revisions in the manuscript: The new experimental results have been added as Supplementary Figure 13 and Supplementary Figure 14 in the revised Supplementary Information with the description of “As different printing speeds require different UV intensities and exposure times of one layer, which leads to different printing times and different ratios of PS particles for the same 3D structure. In detail, higher printing speed under higher UV intensity will lead to larger dry volume of polymer skeleton (Supplementary Figure 13) and a smaller ratio of PS latex particles solidified inside the skeleton, resulting in a larger particle spacing (Supplementary Figure 14) and a red-shifted structural color.” in Line 5, Page 12 of the revised manuscript.

7. On page 8, the author thinks that “...PS latex particles inside the polymer skeleton is uncontrollable without the suction force under discontinuous printing process...”. Then, in the printing of segmental koi fish (Figure 4), did the printing process need to be paused to change the colloidal inks? What’s the influence towards the PC structure?

Reply: Thanks for the reviewer’s comments.

Figure R7 (Fig. 4k in the revised manuscript). Surface SEM image of interface between the two adjacent segments. The two sides of the interface are printed from the PS latex particle diameters of 192 nm and 214 nm, respectively.

As the reviewer mentioned, the printing process needs to be paused for a few seconds to change another colloidal ink in situ with different particle diameters for printing the next segment, where the just printed structure serves as the new supporting plate for the continuous printing of the next segment. The stop may lead to a non-close-packed assembly of latex particles between the two adjacent segments. As the different structural color inks contain the same UV-curable system, the interface between the two neighboring segments can be well connected by the polymer skeleton and the gap dimension is tiny (**Figure R7**), which has little influence on the fidelity and precision of the printed PCs structure.

8. The reflection peaks in Figure 4n was questionable. If the spectra were measured by a general optical fiber with collecting diameters of several millimeters, the reflection peak should be a doublet peak according to the size of the fish. If the spectra were measured by a microscale optical probe, the peak II, V, and VI actually reflected a transitional but still uniform lattice spacing between the neighboring PCs. Experimental proofs were required to support the reflection results.

Reply: Thanks for the reviewer’s comments. The reason of a single reflectance peak at the boundary, instead of two peaks, is ascribed to the detection method of the reflectance spectra, which displays a composite color rather than two separate colors. As the interface is sharp and is on the microscale, we measure the reflectance spectra by the fibre optic spectrometer (NoVA-EX, IdeaOptics, China) through an optical microscope (BX53, OLYMPUS, Japan). The detection diameter is related to the magnification of objective lens, which is usually tens of micrometers. In our experiment, the reflectance spectra of interface between the parts printed from particle diameters of 192 nm and 214 nm are measured at magnification of 50X, whose detection diameter is $\sim 10 \mu\text{m}$. In addition, we also measure the same interface from the lens of 20X with detection diameter of $\sim 25 \mu\text{m}$. As shown in **Figure R8**, the reflectance spectrum displays one stopband, whose position is the same as the one characterized from 50X lens, confirming that the detection range of optical lens in the microscale displays the composite color rather than two single colors.

Figure R8. The reflection spectra measurement under different magnifications of the interface between two adjacent parts.

Revisions in the manuscript: We have added the reflectance characterization method with the description of “Reflectance spectra are measured by fibre optic spectrometer (NoVA-EX, IdeaOptics, China) through an optical microscope (BX53, OLYMPUS, Japan).” in the “**Characterization**” part of **Method** section of the revised manuscript.

References

1. Liao, J. *et al.* 3D-printable colloidal photonic crystals. *Mater. Today* **56**, 29-41 (2022).
2. Parker, S. T. *et al.* Biocompatible Silk Printed Optical Waveguides. *Adv. Mater.* **21**, 2411-2415 (2009).

3. Lorang, D. J. et al. Photocurable Liquid Core-Fugitive Shell Printing of Optical Waveguides. *Adv. Mater.* **23**, 5055-5058 (2011).
4. Frascella, F. et al. Three-Dimensional Printed Photoluminescent Polymeric Waveguides. *ACS Appl. Mater. Interfaces* **10**, 39319-39326 (2018).
5. Nizamoglu, S. et al. Bioabsorbable polymer optical waveguides for deep-tissue photomedicine. *Nat. Commun.* **7**, 10374 (2016).
6. Choi, M. et al. Light-guiding hydrogels for cell-based sensing and optogenetic synthesis in vivo. *Nat. Photonics* **7**, 987-994 (2013).
7. Li, K. et al. Facile full-color printing with a single transparent ink. *Sci. Adv.* **7**, eabh1992 (2021).
8. Patel, B. B. et al. Tunable structural color of bottlebrush block copolymers through direct-write 3D printing from solution. *Sci. Adv.* **6**, eaaz7202 (2020).
9. Boyle, B. M., French, T. A., Pearson, R. M., McCarthy, B. G. & Miyake G. M. Structural color for additive manufacturing: 3D-printed photonic crystals from block copolymers. *ACS Nano* **3**, 3052-3058 (2017).
10. Liu, Y. et al. Structural color three-dimensional printing by shrinking photonic crystals. *Nat. Commun.* **10**, 4340 (2019).
11. Wang, K. et al. 3D Printing of Viscoelastic Suspensions via Digital Light Synthesis for Tough Nanoparticle-Elastomer Composites. *Adv. Mater.* **32**, e2001646 (2020).
12. Tumbleston, J. R. et al. Continuous liquid interface production of 3D objects. *Science* **347**, 1349-1352 (2015).
13. Zhang, Y. et al. Super-Elastic Magnetic Structural Color Hydrogels. *Small* **15**, e1902198 (2019).
14. Zhao, Z. et al. Bioinspired Heterogeneous Structural Color Stripes from Capillaries. *Adv. Mater.* **29**, 1704569 (2017).
15. Fu, F., Shang, L., Chen, Z., Yu, Y. & Zhao, Y. Bioinspired living structural color hydrogels. *Sci. Robot.* **3**, eaar8580 (2018)
16. Wu, L. et al. Bioinspired Ultra-Low Adhesive Energy Interface for Continuous 3D Printing: Reducing Curing Induced Adhesion. *Research* **2018**, 4795604 (2018).
17. Zhang, Y. et al. Continuous 3D printing from one single droplet. *Nat. Commun.* **11**, 4685 (2020).

Reviewers' Comments:

Reviewer #1:

Remarks to the Author:

The revised manuscript has been improved by addressing the comments from the referees. There are some points that are still not clearly discussed.

1. Do the surface images in Figure R3 show the particle arrangement in the planes parallel to the UV window of the DLP machine? If so, (111) planes of a fcc lattice are produced along the horizontal cross-sections of printed structures. Then, how particles are arranged along the side surfaces of the printed structures exposed to air? The crystal orientation on the side surfaces is important factor to determine the structural colors.
2. It seems that particles are randomly arranged in the inner part even after the complete evaporation, as shown in the lower image of Figure R3A and suspected from broad reflectance peaks. It is suggested to discuss the internal arrangement of particles on the surface and interior. It is important to characterize the depth of crystalline skin from the surface, which will determine the reflectivity.
3. The amorphous and crystalline nanostructures in the printed objects before and after the water evaporation in Figures R3D and A, respectively, may indicate that the evaporation induces crystallization in a way of evaporation-induced self-assembly (EISA)? It is suggested to discuss the effect of evaporation toward crystallization in this work.
4. It seems that boundary between two-different color regions is clearly defined in Figure 4k. But, there is a single peak in the spectrum at the wavelength between two. It is still difficult to understand from what the authors explained in the revision.

Reviewer #2:

Remarks to the Author:

In the revised manuscript, the authors have addressed my concerns fairly well. The improvement of current work compared to the report Mater. Today has been clarified. Many experimental results, such as the influence of printing speed on reflection signals, now has been explained in a deeper way. I would like to suggest it publication on Nat Commun after the following minor problems.

In the reply to the 8th question of reviewer #2, the authors have explained why the normalized reflection of region II, V, and VI shows a single peak instead of a double peak. Although the measurement is true, because the reflection peak is too broad to present two individual peaks with different wavelengths, the current results in Figure 4n still pass a confusing information to the readers, as the mixing colors are usually hard to be related to a sharp single peak. Looking back to the 4th suggestion, such problem was exactly caused by the normalization of the reflection signal. Without normalization, a weaker and broader reflection peak of these mixing regions compared to the monochromatic region will explain the color distribution fairly well. Untreated reflection signal should be supplied in Fig 3, and 4. The photo in Fig 3j might be removed to make the images clear.

Responses to Reviewer # 1

The revised manuscript has been improved by addressing the comments from the referees. There are some points that are still not clearly discussed.

Reply: We greatly appreciate the reviewer for the positive assessment and valuable suggestions. According to the reviewer's comments, the manuscript has been carefully revised. We hope that the revised manuscript would be suitable for publication in *Nature Communications*.

1. Do the surface images in Figure R3 show the particle arrangement in the planes parallel to the UV window of the DLP machine? If so, (111) planes of a fcc lattice are produced along the horizontal cross-sections of printed structures. Then, how particles are arranged along the side surfaces of the printed structures exposed to air? The crystal orientation on the side surfaces is important factor to determine the structural colors.

Reply: Thanks for the reviewer's comments. The "Surface" SEM images in the mentioned Figure R3 (the upper row) characterized the side surface of the 3D printed slab structure after complete evaporation, as displayed as the dotted rectangle in Fig. 2c, which is first immersed inside the resin droplet and is then exposed to air after continuously lifting of the supporting plate. The "Cross Section" SEM images in the mentioned Figure R3 (the lower row) characterized the horizontal cross-sections of slab structure, which is parallel to the UV window of the DLP machine as the reviewer mentioned. They are selected as located in the middle of dotted rectangle of Fig. 2c and are perpendicular to the surface SEM images. Both the "Cross section" and the "Surface" SEM images display hexagonal assembly. As the printing process is based on the continuous layer-by-layer printing manner, the stacking of the margins of the cross-sectional images (the lower row of the mentioned Figure R3) along the moving direction of the supporting plate in fact comprises the side surface SEM images (the upper row of the mentioned Figure R3).

Figure R1 (Supplementary Figure 5a-h in the revised Supplementary Information). Characterization of the assembly of latex particles on the side surfaces of the printed structure. (A) Scheme of the SEM characterization positions. I-VI

represent the six different surfaces of the 3D printed slab structure. (B-G) SEM images of the I-VI surfaces in (A). (H) Reflectance spectra of the I-VI surfaces in (A).

In order to clearly characterize the crystal orientation of the latex particles along the side surfaces exposed to air, SEM characterization and reflectance spectra measurement on the six surfaces (**Figure R1A**) of the slab structure (10 mm in length and height, 500 μm in width) are performed. Hexagonal assembly are formed throughout all surfaces (**Figure R1B-G**), and consistent characteristic peak positions are obtained on the all six surfaces (**Figure R1H**), which is consistent with the displayed color in Figure 2c.

Revisions in the manuscript: Corresponding images have been added as Supplementary Figure 5a-h in the revised Supplementary Information with the description of “After water complete evaporation, the PS latex particles form a close-packed hexagonal arrangement and the polymer chains uniformly intersperse the gaps among the PS latex particles on all side surfaces and throughout the entire cross section (Fig. 2f, Supplementary Figure 5)” in Line 6, Page 8 of the revised manuscript.

2. It seems that particles are randomly arranged in the inner part even after the complete evaporation, as shown in the lower image of Figure R3A and suspected from broad reflectance peaks. It is suggested to discuss the internal arrangement of particles on the surface and interior. It is important to characterize the depth of crystalline skin from the surface, which will determine the reflectivity.

Reply: Thanks for the reviewer’s suggestions. For the assembly of the inner part of the printed structure after complete evaporation, the sample for the “Cross Section” characterization is prepared through the brittle fracture method after being frozen by liquid nitrogen, which leads to the random appearance of the upper layer of latex particles or the random missing of the same layer of latex particles during the brittle fracture process. This phenomenon can be ascribed to the flexibility of the hydrogel and the adhesion between the latex particles and the hydrogel. The simultaneous appearance of different layers of latex particles may lead the reviewer to conclude that “It seems that particles are randomly arranged in the inner part even after the complete evaporation”. However, along with the SEM characterization of the printed structure after removing the latex particles (the lower image of the mentioned Figure R3B), it can be considered as hexagonal assembly.

In addition, cross-sectional SEM characterization from the surface to the interior (I to III in **Figure R2A**) of the slab structure and corresponding SEM characterization after selectively removing the PS latex particles are conducted to illustrate the position-related internal arrangement of the latex particles mentioned by the reviewer. As shown in **Figure R2B-D**, the latex particles at different positions display the same tendency of hexagonal arrangement, which is consistent with the image mentioned by the reviewer. In addition, after selectively removing the PS latex particles, the residual polymer skeleton shows arrayed cilia morphology that replicates the vertical gaps among the close-packed particles (**Figure R2E-G**), which further indicates the

hexagonal assembly along the entire cross section.

Figure R2 (Supplementary Figure 5i-o in the revised Supplementary Information). Cross-sectional SEM characterization from the surface to the interior of the slab structure after complete evaporation. (A) Cross-sectional SEM image of the printed slab structure. I-III represent the different SEM characterization positions from the surface to the interior. (B-D) Cross-sectional SEM images of the different positions in (A). (E-G) Cross-sectional SEM images of the different positions in (A) after selectively removing the PS latex particles.

Revisions in the manuscript: Corresponding images have been added as Supplementary Figure 5i-o in the revised Supplementary Information with the description of “After water complete evaporation, the PS latex particles form a close-packed hexagonal arrangement and the polymer chains uniformly intersperse the gaps among the PS latex particles on all side surfaces and throughout the entire cross section (Fig. 2f, Supplementary Figure 5)” in Line 6, Page 8 of the revised manuscript.

3. The amorphous and crystalline nanostructures in the printed objects before and after the water evaporation in Figures R3D and A, respectively, may indicate that the evaporation induces crystallization in a way of evaporation-induced self-assembly (EISA)? It is suggested to discuss the effect of evaporation toward crystallization in this work.

Reply: Thanks for the reviewer’s comments. The function of evaporation has been displayed in Fig. 2 with corresponding descriptions. It is different from the simple evaporation-induced self-assembly, as the particles arrange along with the polymerization and assemble inside the confined polymer skeleton in this system, which had been illustrated with corresponding descriptions in Fig. 2d-f and Fig. 2j.

4. It seems that boundary between two-different color regions is clearly defined in Figure 4k. But, there is a single peak in the spectrum at the wavelength between two. It is still difficult to understand from what the authors explained in the revision.

Reply: Thanks for the reviewer’s comments. The reflectance spectra without normalization are displayed in **Figure R3**, which still cannot present two individual

peaks. As mentioned by Reviewer #2, the reflection peak at the boundary between two adjacent parts is too broad to represent two individual peaks with different wavelengths, which may be limited by the resolution of the test equipment and the detection range of optical lens in the microscale. As the highlights of our work is the realization of volumetric color property and simultaneous macroscopic printing and microscopic particle assembly via continuous digital light processing 3D printing method, to make it clearer for readers, the reflection spectra of II, IV and VI in Fig. 4n have been removed in the revised manuscript.

Figure R3. The reflectance spectra of the 3D printed colorful koi fish structure without normalization.

Revisions in the manuscript: The positions of interfaces II, IV, and VI in Fig. 4b have been deleted (Figure R4A), and the untreated reflection signals (Figure R4B) have been supplied in Fig. 4n.

Figure R4. The characterization of the 3D printed colorful koi fish structure. (A, Fig. 4b in the revised manuscript) Optical image of the multi-structural color koi fish continuously printed with four different PS latex particle diameters. The dashed curves represent the interface of two segments printed with different PS latex particle diameters. I-IV are the segments printed from the PS latex particle diameters of 192 nm, 214 nm, 230 nm and 265 nm, respectively. The inset on the bottom left is the 3D model of koi fish. (B, Fig. 4n in the revised manuscript) The reflectance spectra of I-IV segments in (A).

Responses to Reviewer # 2

In the revised manuscript, the authors have addressed my concerns fairly well. The improvement of current work compared to the report Mater. Today has been clarified. Many experimental results, such as the influence of printing speed on reflection signals, now has been explained in a deeper way. I would like to suggest it publication on Nat Commun after the following minor problems.

Reply: We thank the reviewer very much for the positive comments and the suggestions for improving the manuscript. According to the reviewer's comments, we have made revisions on our manuscript. The suggestions improve the quality of the revised manuscript. We hope that the revised manuscript would be suitable for publication in *Nature Communications*.

In the reply to the 8th question of reviewer #2, the authors have explained why the normalized reflection of region II, V, and VI shows a single peak instead of a double peak. Although the measurement is true, because the reflection peak is too broad to present two individual peaks with different wavelengths, the current results in Figure 4n still pass a confusing information to the readers, as the mixing colors are usually hard to be related to a sharp single peak. Looking back to the 4th suggestion, such problem was exactly caused by the normalization of the reflection signal. Without normalization, a weaker and broader reflection peak of these mixing regions compared to the monochromatic region will explain the color distribution fairly well. Untreated reflection signal should be supplied in Fig 3, and 4. The photo in Fig 3j might be removed to make the images clear.

Reply: Thanks for the reviewer's suggestions. Accordingly, we have supplied the reflectance spectra without normalization as displayed **Figure R3**.

Figure R3. The reflectance spectra of the 3D printed colorful koi fish structure without normalization.

Considering that the results are related with the resolution of the test equipment and the detection range of the optical lens, and the highlights of our work is the realization of volumetric color property and simultaneous macroscopic printing and microscopic particle assembly via continuous digital light processing 3D printing method. So, in

order to make it clearer to the readers, the reflection spectra of II, IV and VI in Fig. 4n have been removed in the revised manuscript. Furthermore, the untreated reflection signals are supplied in Fig. 3 and 4, and the photo in Fig. 3j is moved to the revised Supplementary Information.

Revisions in the manuscript: The positions of interfaces II, IV, and VI in Fig. 4b have been deleted (Figure R5A), and the untreated reflection signals (Figure R5B) have been supplied in Fig. 4n. In addition, Fig. 3j has also been revised with the untreated reflection signals (Figure R5C), with the movement of the inset photo (Figure R5D) to the revised Supplementary Information as Supplementary Figure 11 with the description of “As displayed in Fig. 3j and Supplementary Figure 11, with a fixed printing speed of 20 $\mu\text{m/s}$, the structural color of the printed slab structure can be regulated from cyan, green-yellow, orange, red-orange to red through increasing the diameter of the PS latex particle from 192 nm, 214 nm, 230 nm, 245 nm to 265 nm without changing other ink components” in Line 5, Page 11 of the revised manuscript.

Figure R5. The optical characterization of the 3D printed structures. (A, **Fig. 4b in the revised manuscript**) Optical image of the multi-structural color koi fish continuously printed with four different PS latex particle diameters. The dashed curves represent the interface of two segments printed with different PS latex particle diameters. I-IV are the segments printed from the PS latex particle diameters of 192 nm, 214 nm, 230 nm and 265 nm, respectively. The inset on the bottom left is the 3D model of koi fish. (B, **Fig. 4n in the revised manuscript**) The reflectance spectra of I-IV segments in (A). (C, **Fig. 3j in the revised manuscript**) The reflectance spectra of the 3D printed structures from PS latex particles with different diameters. (D, **Supplementary Figure 11 in the revised Supplementary Information**) Optical image of 3D printed structure with different structural colors. From left to right, the PS latex particle diameters used are 192 nm, 214 nm, 230 nm, 245 nm, and 265 nm, respectively.

Reviewers' Comments:

Reviewer #1:

Remarks to the Author:

The authors have properly addressed my comments and I have no objection for the publication of this work as is.

Reviewer #2:

Remarks to the Author:

The authors have revised the manuscript according to the suggestions. I am happy to recommend its publication on Nat. Commun. in its current form.